# Beyond Prompts: Dynamic Conversational Benchmarking of Large Language Models

**David Castillo-Bolado**\*, **Joseph Davidson**\*, **Finlay Gray**, **Marek Rosa**

GoodAI
{david.castillo,joseph.davidson}@goodai.com

## Abstract

We introduce a dynamic benchmarking system for conversational agents that evaluates their performance through a single, simulated, and lengthy user↔agent interaction. The interaction is a conversation between the user and agent, where multiple tasks are introduced and then undertaken concurrently. We context switch regularly to interleave the tasks, which constructs a realistic testing scenario in which we assess the Long-Term Memory, Continual Learning, and Information Integration capabilities of the agents. Results from both proprietary and open-source Large-Language Models show that LLMs in general perform well on single-task interactions, but they struggle on the same tasks when they are interleaved. Notably, short-context LLMs supplemented with an LTM system perform as well as or better than those with larger contexts. Our benchmark suggests that there are other challenges for LLMs responding to more natural interactions that contemporary benchmarks have heretofore not been able to capture.

## 1 Introduction

The capabilities of Large Language Models (LLMs) have been primarily evaluated through isolated tests (see §6), focusing on increasing data volumes and context size (c.f. [Su et al., 2024, Chen et al., 2023, Peng et al., 2024, Zhang et al., 2024a]). However, these tests are single shot and single topic (Figure 1), and therefore often overlook the most common user interaction mode: chat-based conversation.

In response, we introduce the LTM Benchmark[1], an automated system designed to evaluate the Long-Term Memory (LTM) and Continual Learning (CL) capabilities of conversational agents. The LTM Benchmark engages agents in a single, prolonged conversation, incorporating multiple tasks and distractions to simulate realistic and meaningful interactions. This approach provides a more comprehensive assessment of an agent's ability to effectively use and integrate information across extended dialogues. Furthermore, we show that this "conversational multitasking" structure significantly degrades the test performance of LLMs, indicating that their real-world capabilities are not fully revealed through most contemporary benchmarks. The LTM benchmark is primarily synthetic, and can be generated to result in conversations of arbitrary length, which may potentially surpass the context size of the LLMs under test.

Our contributions can be described thus:

- An automatic benchmarking system that evaluates an agent by conversing with it, interleaving all tests in a single conversation.

---

[1]All data, experiments, and code for implementing, and running the LTM benchmark mentioned in this paper are under an open source license and available, in full, at: https://github.com/GoodAI/goodai-ltm-benchmark.

38th Conference on Neural Information Processing Systems (NeurIPS 2024) Track on Datasets and Benchmarks.

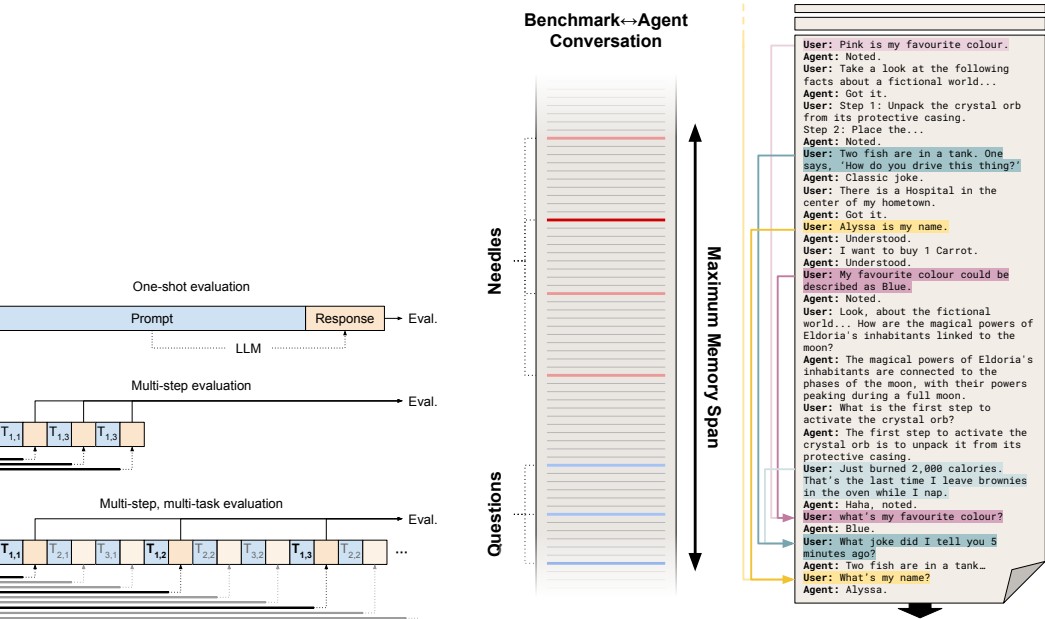

Figure 1: Standard one-shot evaluations focus on building a challenging prompt and evaluating the LLM's response. In a conversation there are many LLM calls involved and the *prompt* grows monotonically, and the task can either stay constant or be switched regularly.

Figure 2: (left) Outline of a test's structure as part of the entire benchmark conversation. Needles and questions are messages that spread out throughout the conversation, aiming to take as much space as the memory span allows. (right) Zooming in, we can see how the tests are intertwined, and how different questions make reference to distinct pieces of information.

- An initial battery of tests evaluating different aspects related to LTM and CL. Both the tests themselves and the generators used to create them.
- An evaluation and analysis of results for some of the most capable language models to date.

## 2 Definitions and Terms

**Needles and Haystack**. Terms introduced by the Needle in a Haystack (NIAH) benchmark [Kamradt, 2024]. Needles are sentences in the context that are relevant to the task, and which are required to be found and retrieved. The term haystack refers to the remaining (usually unrelated) text, that acts as a spacer or distractor, depending on the complexity of the *hay*. In this paper, we use "needles" to refer to all sentences that contain information relevant to the task.

**Information Integration**. The ability to take multiple related needles from different locations (either context, or some other memory system) and integrate them into a complete and consistent view. This often implies a continual and iterative process of revision of past knowledge, in the search for novel insights or conclusions.

**Memory Span**. The amount of information that a model is able to take in order to produce a response. In the case of LLMs, this is upper bounded by their maximum context sizes, which determines how much text they can take as input. We also employ this term to refer to how much space in the conversation a specific test uses. For example, we may say that the memory span for a test is 120k tokens, which means that the tested agent will find all necessary information within the latest 120k tokens of the conversation.

**Long-Term Memory (LTM)**. The ability to recall and effectively utilize past information. LTM encompasses several skills related to the generation and management of memories, which include but are not limited to recall, information integration, and the handling of conflicting information. This broad definition acknowledges that different implementations of systems or agents may exhibit such ability in different degrees. Current implementations of LTM commonly rely on an LLM as

the main controller, a set of auxiliary components for the management of memories (e.g. scratch-pads and semantic vector databases) and some sort of Retrieval Augmented Generation [Huang and Huang, 2024].

# 3 The LTM benchmark

The LTM Benchmark is a means for evaluating the memory and information integration capabilities of an agent via a single written conversation. It does not make any assumptions about the underlying technology or implementation of the agents being tested. Instead, the focus is on benchmarking the agent's skills from a purely functional perspective, setting only two broad requirements: 1.) The agent must be able to communicate in written English, and 2.) The agent must send a message exclusively when prompted, providing a single response message to each message received.

By following this approach we aim to contribute an objective evaluation tool, capable of assessing the capabilities of current and future agents, and pointing to the most relevant challenges towards the next generation of conversational agents.

Over the course of the test interaction, the agent is presented with various pieces of information at different points in time, and is asked questions or challenged with situations that rely, directly or indirectly, on this information. We have equipped this benchmark with an initial battery of tests which reflect currently challenging skills, but we expect the benchmark to evolve according to the needs of the research community, discarding tests that become trivial and incorporating new ones that evaluate newly identified and more challenging capabilities.

As a mostly synthetic benchmark, the generation processes is both transparent and reproducible. As such, we have released all benchmark results and generators. The benchmark is dynamic, but all generation and scheduling processes are deterministic. The one source of nondeterminism in the testing is from the agents themselves. The length of agents' replies will have knock-on effects in the amount of filler tokens that are used, and that in turn may influence the replies of the agent. More details on this and our mitigation strategies can be found in Appendix C.

## 3.1 Test structure

Every test in the benchmark is automated and takes place as part of the same conversation. Unlike other LLM-oriented benchmarks, our benchmarking system does not isolate tests as separate prompts, but intertwines them in a continuous exchange of messages between the agent and user. This approach aims to keep the conversation as natural as possible, as we hypothesise that the results obtained in this way will be more likely to generalize to chat-based interactions.

The conversation begins with a message explaining the overall situation to the agent, and continues with all tests' messages and agent responses (see Figure 3). Every test encompasses a series of interventions from the benchmark system (taking the user's role), which are uniformly distributed across a previously-delimited area in the conversation known as the memory span of the test (see Figure 2). The test's span can be set independently from the agent's effective input size, which allows for tests surpassing current LLMs' context lengths. Other constraints like token and time waits (see episodic memory and *Jokes* scenario in §3.2) also form part of the test information, which is provided to the benchmark's scheduling system.

To disperse test messages, the benchmark system uses distraction messages, which are messages from other unrelated tasks. These unrelated tasks are preferentially other tests' messages, which greatly improves the benchmark's efficiency, but sometimes it is just not possible to make use of a message from another test and we have to rely on dummy tasks. The dummy task that we rely on is the extraction of answers from the TriviaQA dataset [Joshi et al., 2017] (Figure 4).

A test can be arbitrarily complex depending on its content and definition. Test's messages are either delivered in a static manner (providing a script) or dynamically (via a functional description), depending on the requirements of the testing scenario. The LTM benchmark's scheduling system coordinates the execution of tests and weaves them together into a seamless conversation, while respecting the memory span, test compatibility constraints and optimizing the use of tokens. Finally, once the last test question is answered, the test is scored according to the agent's performance. The complexity of this scoring mechanism can vary greatly too, from a simple pattern matching

```
Tester: I am going to subject you to a Long-Term Memory benchmark.  In the
following, I will be giving you different kinds of information and I expect
you to answer extremely briefly, only providing the responses that you are
required to provide.  Otherwise, provide just short confirmations.
Understood?
Agent: Understood.
Tester: Please add 2 Toothpaste to my shopping list.
Agent: Added!
Tester: Start calling me by my name which is Liam.
...
```

Figure 3: Example beginning of a benchmark. The benchmark puts the agent in situation and then follows by interleaving test messages.

```
Tester: Here are some trivia questions and answers for you to process.  Please
extract all of the answers in json form as a single message:  E.g ["answer
1", "answer 2", ...]
Q: Of which Saxon kingdom was Offa a King?, A: MERCIA
Q: What is the name of the test cricket venue in Manchester, England?, A: Old
Trafford
...
```

Figure 4: Example of the dummy task based on the TriviaQA dataset.

statement to an elaborate evaluation protocol using LLMs as evaluators. Appendix A contains more details relating to the content and evaluations methods of the tests.

## 3.2   Test scenarios

We have included an initial set of tests that evaluate a series of memory and learning skills. Inspired by research in psychology and cognitive science, we have built meaningful and organic test scenarios, which abstract away the agent's implementation and focus on the agent's abilities to recall and effectively leverage past information. All test scenarios evaluate a subset of the following skills:

- **Recall (R).** Simple retrieval of past information, usually based on content matching.
- **Conflicting information (C).** Managing of information that contradicts existing knowledge or previously acquired information.
- **Episodic memory (E).** Addressing of memories based on temporal information.
- **Spatial memory (S).** Association of memories to positions in some space.
- **Prospective memory (P).** Memories that address future or hypothetical situations.
- **Theory of mind (T).** Memories pertaining to a third persons' thoughts or feelings.
- **Information integration (I).** The act of combining different pieces of information in order to construct a complex or more abstract overall picture.

And in order to assess these skills, we have built the following test scenarios:

- **Colours (R)**: We state our favourite colour several times and in different ways, changing preferences every time. Finally, we ask the agent what our favourite colour is.
- **Name list (R, C)**: We change names multiple times, then instruct the agent to recall the complete list of names that we have gone by.
- **Jokes (E)**: We tell the agent a series of jokes at different and spaced points in time, the agent is then asked to recall the joke told to it a given number of hours and minutes ago.
- **Locations directions (S, I)**: We give the agent a series of locations from a fictional town. Each statement places a new location into the town relative to the previous one. Finally, the agent is asked how to get from the first place to the last.

- **Quotes (P)**: The agent is given a quote from a well-known author, and then it is asked to append the quote to its $n^{th}$ response.

- **Trigger response (C, P)**: The agent is given a particular trigger phrase, and instructed on how it should respond to it. We later use those trigger phrases and see if the agent responds appropriately.

- **Sally–Anne (T)**: The agent is evaluated with scenarios from the ToM QA Dataset [Nematzadeh et al., 2018], which assess the ability of the agent to acknowledge the beliefs of another subject (theory of mind).

- **Spy meeting (R, I)**: The agent is contacted by three different people, each of them giving cryptic messages related to where, when, and what to bring to a clandestine meeting. Finally, we ask the agent to retrieve and interpret those messages.

- **Shopping list (R, C, I)**: Over the course of the conversation, the agent receives a series of updates regarding the user's shopping list. The agent is then asked about the status of the shopping list, which has suffered several changes by that time.

- **ChapterBreak (potentially any)**: This dataset [Sun et al., 2022] presents the agent with up to 10 chapters from a book, and 6 options for the following chapter beginning. It is a known fact that $\approx 60\%$ of the samples in the ChapterBreak dataset have no solution, so we have manually selected 4 samples for which we have made sure that finding a solution is possible.

- **Restaurant (C, I)**: The agent is placed in an everyday situation involving sitting at a restaurant, ordering food and dealing with a series of unfortunate events. This highly dynamic test evaluates the coherence of the agent's actions during the experience. Among the evaluated points, this test scores positively ordering items that are in the menu and speaking up when the waiter brings the wrong dish.

These scenarios are tested multiple times in the conversation, for which the benchmark system instructs the agent to forget or disregard the information in the previous test. This approach implies that every additional repetition may require the agent to deal with an increasing amount of conflicting information.

### 3.3 Generation processes

This benchmark is considered mainly synthetic because of how the data is presented, where tasks are interwoven with other tasks and previous agent responses, and not so much because the data itself being synthetic. Some of the test scenarios do not rely on synthetic data, but on already existing –and sometimes relatively small– datasets. It is the interleaving of tasks and messages what makes each interaction unique, since the conversation history is highly diverse and sensitive to the agent's responses.

Each test scenario has its own generation method, which is conditioned on the global seed as well as the index of the test repetition. All data required for running the tests is generated upfront during initialization and saved to disk. Dynamic tests also rely on seeded random generators and zero-temperature LLM calls to synthesize data and react to the agent's responses. We refer the reader to Appendix A for a detailed description of the generation of each scenario included in this edition of the LTM benchmark.

Besides this initial data generation process, the benchmark relies on a scheduling system to coordinate the intertwined delivery of test messages while attending to the different tests' requirements and other constraints. In order to accomplish that, each test is required to communicate those constraints along with every message delivered: namely, how much time it should pass or how many tokens should be exchanged before its next intervention, if any at all. This is similar to how yielding mechanisms work in computational multithreading. The policy of this message scheduling system can be summarized by the following rules:

- Incompatible tests cannot run concurrently.

- Messages from a same test are delivered until the test ends or yields.

- When a test yields or ends, the system gives priority to resuming other waiting tests before starting a new one.

- Only if no test can be resumed or started, the system will try to unblock the situation by sending dummy tasks (using tokens) or performing time jumps (using up time).

A detailed specification of the benchmark's scheduling system and policy can be found in Appendix C.

### 3.4 LTM Implementations

In these tests, we test both unmodified LLMs, as well as LLMs augmented with LTM systems. These LTM implementations fall into broadly two categories: those that implement memory through modifications of the LLM, and those that implement memory through modifications to the context of the LLM, but not through architectural choices.

Works such as LongMem [Wang et al., 2023] capture the hidden states of the transformer and use those residuals as input to a "SideNet" which can attend to a cached memory of inputs or documents. Memorizing Transformers [Wu et al., 2022] augment one of the final layers of a transformer model with a $k$-nn search over hidden states saved to a stack. MemoryLLM [Wang et al., 2024] is another model which maintains a pool of updatable parameters within the latent space of the transformer.

Works such as MemoryBank [Zhong et al., 2024] and MemGPT [Packer et al., 2023] instead develop memory as a tool which augments the input context of the LLM, rather than influencing token generation directly. These approaches allow for some flexibility in their choice of model, and permit the use of commercial LLMs such as Claude or GPT.

Agents with LTMs can be further distinguished by whether they use their LTM *actively* or *passively*. In an active system, the LLM has some awareness of the functionality of the LTM, and can choose how to use it. This often takes the form of a multi-step resolution to a user query by making decisions such as when memory should be read from or written to. Passive systems on the other hand perform reads and writes automatically in a typically fixed loop.

With the aim of offering some reference results for agents with LTM, in our tests we include results from our baseline LTM system, as well as from MemoryBank and MemGPT. Other models were not included due to a lack of pretrained weights or a reliance on early completion LLMs, which are not instruction fine-tuned and cannot withstand the required conversational interaction.

MemoryBank places the focus on empathy and psychological companionship, and stores conversation and summaries of those conversations in chronological order, along with a summary of the user. This information is kept in a vector database, and is passively queried during its conversation with the user. MemoryBank also implements a human-like memory forgetting curve.

An example of an active system is MemGPT. This agent attempts to answer queries by consulting multiple forms of memory. It contains an in-context *core* memory, and external *archival* and *recall* memories. The core memories are always visible as a part of the context, whereas the archival and recall memories, need to be queried and are for storing long running events or documents, and the chat history respectively. The agent can update its archival or core memories via function calls.

Our baseline LTM system[2] uses both a vector database and a JSON scratchpad. On receiving a user message, the agent generates multiple queries for the vector database and retrieves chunks that are not currently part of the context. The scratchpad is updated (or not) after the LLM generates a response to the query, and is similar to MemGPT's core memory in that it is always present in the context.

## 4 Results

We have produced a standard configuration of the LTM benchmark and used it to evaluate and compare some of the current leading LLMs, in addition to a series of LTM-enabled agents. For the configuration, we have included three repetitions of each scenario, and we have set the number of needles for the scenarios requiring it: in *Colours*, three changes; in *Name List*, five different names; in *Jokes*, four jokes; in *Locations Directions*, the path goes through six locations; and in *Shopping List* the list updates six times. We run the same benchmark with a number of different

---

[2]More details of the LTM system can be found on GitHub at: https://github.com/GoodAI/goodai-ltm.

Table 1: Benchmark results for distinct agent configurations: open-source LLMs, commercial LLMs, and LTM agents. *Context* stands for the maximum input tokens available to the LLM. All tests are out of 11 points, apart from the 2k benchmark, which is out of 10. Underscored values are the result of two benchmark runs. A visual representation of these results can be seen in Figure 5.

| Model | Context | Isolated | | 2k | | 32k | | 120k | | 200k | | 500k | |
|---|---|---|---|---|---|---|---|---|---|---|---|---|---|
| | | Score | std | Score | std | Score | std | Score | std | Score | std | Score | std |
| Mixtral 8x7B | 32000 | 5.0 | 0.9 | 1.4 | 0.5 | 0.1 | 0.1 | 0.1 | 0.1 | 0.1 | 0.1 | 0.1 | 0.1 |
| Mixtral 8x22B | 65536 | 4.9 | 1.0 | 5.6 | 0.7 | 0.0 | 0.0 | 0.1 | 0.1 | 0.1 | 0.1 | 0.1 | 0.1 |
| Llama 3 70B | 8000 | 8.2 | 0.7 | 1.9 | 1.0 | 0.2 | 0.0 | 0.2 | 0.0 | 0.2 | 0.0 | 0.2 | 0.0 |
| GPT-3.5 turbo | 16384 | 4.1 | 0.8 | 4.7 | 0.8 | 0.1 | 0.1 | 0.0 | 0.0 | 0.0 | 0.0 | 0.0 | 0.0 |
| GPT-4 turbo | 128000 | 7.9 | 0.1 | 6.6 | 0.7 | _5.2_ | _1.3_ | 5.8 | 1.0 | 3.9 | 0.9 | 1.0 | 0.5 |
| GPT-4o | 128000 | 7.6 | 0.9 | 5.9 | 0.7 | _5.6_ | _1.4_ | _5.9_ | _1.1_ | 5.2 | 0.9 | 0.9 | 0.5 |
| GPT-4o-mini | 128000 | _7.8_ | _0.7_ | _5.6_ | _1.0_ | _4.1_ | _0.7_ | _5.1_ | _1.2_ | _4.8_ | _0.8_ | _1.4_ | _0.6_ |
| Claude 3 Opus | 200000 | 8.3 | 0.8 | **7.8** | **0.9** | 6.7 | 1.0 | 7.4 | 1.1 | 5.1 | 0.9 | 3.4 | 0.4 |
| Gemini 1.5 Pro | 1000000 | 7.4 | 0.9 | 6.5 | 0.8 | 6.4 | 0.8 | 7.0 | 0.5 | _**7.7**_ | _**1.0**_ | _**6.1**_ | _**1.3**_ |
| LTM Claude 3 Opus | 16384 | 8.7 | 0.9 | 7.5 | 0.7 | 5.0 | 1.0 | 5.7 | 1.2 | 6.4 | 1.1 | 4.9 | 1.0 |
| LTM GPT-4 turbo | 16384 | **9.2** | **0.5** | 6.3 | 0.8 | 5.2 | 0.9 | 5.0 | 1.0 | 5.3 | 0.9 | 3.1 | 0.8 |
| LTM GPT-4o-mini | 16384 | _8.1_ | _0.7_ | _5.4_ | _0.8_ | _4.7_ | _1.0_ | _4.6_ | _1.1_ | _4.2_ | _0.9_ | _4.7_ | _1.0_ |
| LTM Llama 3 70B | 8000 | 8.4 | 0.8 | 6.9 | 1.0 | 5.0 | 0.9 | 4.7 | 1.0 | 5.6 | 0.7 | 4.8 | 0.8 |
| MemGPT GPT-4o-mini | 16384 - 40000 | _6.8_ | _0.9_ | _4.8_ | _0.7_ | _2.3_ | _0.7_ | _3.2_ | _0.8_ | _3.6_ | _0.8_ | _2.0_ | _0.7_ |
| MemoryBank GPT-4o-mini | 16384 - 60000 | 6.4 | 0.7 | 3.5 | 0.7 | 2.0 | 0.5 | 2.8 | 0.5 | 3.3 | 0.8 | 3.6 | 0.6 |

memory spans: 2k, 32k, 120k, 200k, and 500k tokens. Additionally, we include results for the same tasks evaluated in isolated scenarios, where there is neither task-switching nor distractors in the conversation (resembling standard benchmarks). We also exclude *ChapterBreak* from the 2k configuration to avoid exceeding the memory span.

A benchmark run consists of 33 tests (11 scenarios and 3 repetitions each), which measure the agent's performance on the different scenarios under diverse contexts. All scores are normalized to $[0, 1]$ and scores from the same scenario are averaged together, which makes for a maximum of 11 points per benchmark. In order to obtain accurate distribution estimates, we generate 1,000 possible outcomes by repeatedly sampling one result from each scenario and adding them together. After running the initial set of benchmarks for all configurations, we allocated the remaining budget to configurations where reducing variance would be most impactful —such as those with high scores and competing closely with others. These additional runs use a different seed and are marked with underscores in Table 1.

The benchmarks are executed using a variety of on-demand APIs from various providers, including Google, OpenAI, and Anthropic. The open source models were tested via TogetherAI's API. Running and evaluating all the benchmarks in this paper cost $\approx$ \$7,200 and took 142 hours.

The tested LTM agents are capable of using different LLMs. On a cost-to-quality ratio, we found that GPT-4o-mini works well. We have tried to keep the context limit for the LTM agents to 16k tokens[3], but the MemoryBank and MemGPT systems cannot artificially restrict their context sizes. We observed that the volume of memories which they retrieve later in the benchmark can balloon the context sizes to around 60k or 40k tokens respectively.

In Table 1 we show the scores achieved by each agent configuration under all memory span settings. We also show a visual representation of the same results in Figure 5. These results show that LLMs perform well on short memory spans, but they struggle as soon as the memory span surpasses their maximum context size, as the relevant information is simply lost. Conversely, LTM agents suffer less when the memory span is increased.

## 5   Analysis

In general, the benchmark has proved to be comprehensive, representing a significant step-up in terms of evaluation results that are robust and hopefully translate to real-world scenarios. One example of this is in GPT-4o, which achieves 100% NIAH coverage and nearly perfect results on the NIAN test [Burns, 2024], yet it scores similarly to GPT-4 turbo in our benchmark. This corroborates OpenAI's claims [OpenAI, 2024] and anecdotal evidence which describes GPT-4o's interactivity to

---

[3]We only use fewer than 16k tokens with Llama 3, which has an input limit of 8k tokens.

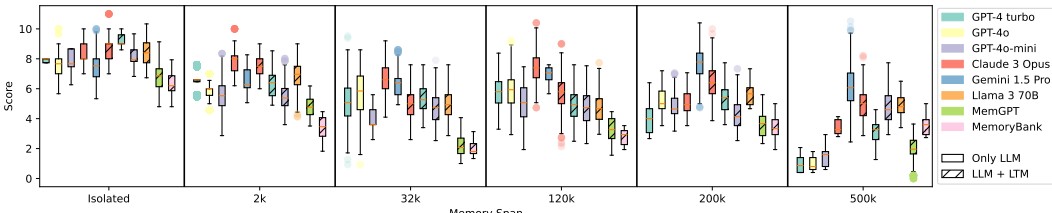

Figure 5: Scores obtained for all agent and memory span configurations. Solid boxes correspond to vanilla LLMs, while strided boxes represent LTM agents. Each color can be associated with a different LLM. Context sizes for each of the agents vary, and are detailed in Table 1.

be better, but its overall capabilities to be in line with GPT-4 turbo [Yan et al., 2024, ArtificialAnalysis, 2024].

We have found the conversational format to be the main challenge for current LLMs (and especially the open source ones), which work better on short and clean prompts. The results for the isolated tests vs. the interleaved 2k tests reveal the interleaving of tasks as a key factor. There is a minimal difference in the presentation of the tests outside of the interleaving, yet performance drops to below half for two of the the open source LLMs. Commercial LLMs are more robust to the interleaved scheme, which may point to how their training is performed. It is worth noting that longer input sizes allow for both the tackling of longer tasks and the presence of more distractors.

Looking at the results in detail (see Appendix D), the Restaurant task was the most variable one. Claude had trouble with the task by being too expressive, adding "stage directions" to its responses. The Mixtral models also often hallucinated interactions between the waiter and diner on their own, resulting in failures. A lot of models that otherwise fail completely on the other benchmark tasks, can get some credit for the first part of the Restaurant task where they are greeted, shown a menu and asked for a drink order in a single message. If the agent plays along with the scenario, it will obtain some score. Our automated evaluation system (powered in some parts by GPT-4 turbo) also fails at some points, like being unable to distinguish between dishes that are discussed, and ones that have been ordered.

Also challenging was the prospective memory tests, where the agents had to append a quote at a future point in time. The most common failure was in the agent not being able to count its statement correctly, usually resulting in an off-by-one error. We have noted that LLMs using special tokens for message delimiters are those which struggle the most in counting them. We believe that this phenomenon might be related to those tokens being rare in the training data.

In the *Locations directions* task, agents were not explicitly asked to visit all locations, but most agents either failed the task or replied with an almost exact replica of the original directions. Notably, merely recalling (and potentially rephrasing) those sentences does not require any kind of spatial thinking.

Traces of pattern-completion behaviour can be seen in all benchmark conversations, in the form of early responses heavily influencing later ones. These later responses usually continue the format and thinking patterns established previously, and an early good response strategy (e.g. listing all items in the shopping list after every update) often translates to all task repetitions succeeding. We have also observed the mimicking of reluctant responses or growing preceding spaces in responses.

Sometimes, and specially with long-context LLMs, mixing information from different repetitions happens. Simple recall does not solve this issue, which requires careful (and ordered) reading of all potentially-relevant information. This seems to imply some sort of iterative processing of the input, which current transformers-based architectures are not very good at. Current LLM architectures are very similar, and yet significant differences can be seen. Such differences mostly originate in the training procedure, but architectural choices can be decisive too. To give some examples, Gemini 1.5 Pro is mostly incapable of solving ToM tasks, but it also exhibits better-than-average spatial reasoning. Llama 3 excels when the prompt is short and clear, and it can beat the best commercial LLMs in this regime, being especially well suited for LTM agents. Mixtral 8x22B also performs exceptionally well in short conversations.

The guardrails on commercial LLMs frequently appear in responses to longer versions of the benchmark tests. Safety measures tend to go off when the tests ask about any kind of personal info or datum that might seem out of the agent's reach, even if the data is somewhere in context. This refrain of "As an AI language model..." could be a default result of the imperfect attention trying to read personal information from the context.

# 6 Other benchmarks

Most LLM benchmarks rely on one-shot evaluations[4] (Figure 1). Prefilled contexts have various token lengths, e.g. up to 18k in LongBench [Bai et al., 2023], $\approx$ 36k in LooGLE [Li et al., 2023], and 200k in InfiniteBENCH [Zhang et al., 2024b]. A common shortcut is to take data points from existing datasets, but it is possible that they are part of the pretraining dataset for a model [Jacovi et al., 2023, Yang et al., 2023]. Datasets like *HotpotQA*, *2WikiMultihopQA*, *Qasper*, and *NarrativeQA* [Yang et al., 2018, Ho et al., 2020, Dasigi et al., 2021, Kočiský et al., 2017] are popular choices for test data and included in LongBench. Mitigation strategies include the use of recent real-world data (LooGLE) and the handcrafting of questions (LongBench). Also based on human-created questions there's L-Eval [An et al., 2023], which aims to bridge the gap between commercial and open-source models.

Needle in a Haystack [Kamradt, 2024] is a common pattern for evaluating LLMs' retrieval skills. Often those using NIAH use their own data and needles, which is an issue because some works have shown that the complexity of the "hay" matters [Pekelis, 2024]: repetitive haystacks or incongruous needles present an easier task for the LLM. More recent variations include the multi-needle retrieval test [Knight-Webb, 2024] and the Needle in the Needlestack (NIAN) test [Burns, 2024]. There are other benchmarks like RULER [Hsieh et al., 2024] and BAMBOO [Dong et al., 2024] which extend the evaluation complexity beyond simple retrieval, although the tests are still comprised of isolated prompts (one-shot). Other long one-shot tests such as Multi-Session Chats [Xu et al., 2022] and ChapterBreak [Sun et al., 2022] evaluate holistic understanding of the context, which entails a potentially high complexity.

There are also evaluation systems intended for agents. AgentBench [Liu et al., 2024], WebArena [Zhou et al., 2023] and Autonomous Replication and Adaptation tests [Kinniment et al., 2023] perform agential evaluations. Similar to us, these benchmarks evaluate the agent within a dynamic environment in which the agent is required to autonomously pursue a goal. AgentSims [Lin et al., 2023] also presents a dynamic environment, but they leave the evaluation open. Bai et al. [2024] incorporate interviews with the agent to some degree, using LLMs as an aid for evaluating the agent's responses. LLF-Bench [Cheng et al., 2023] features a dynamic environment for assessing the continual learning skills of an agent, adapting the original proposal of Mikolov et al. [2018] to an LLM interface.

# 7 Limitations and future work

Despite the interesting results, there are a number of limitations that require further development. This benchmark is also intended to be a living benchmark, which calls for continued work in updating the tests for targeting new skills.

Agents' robustness in this paper are based on only three repetitions, which is not very reliable. The benchmark has a high time and financial cost to running it, and therefore multiple repeats have proven currently infeasible, but we expect this to change in the near future.

Given the breadth of possible responses to any question, the automatic evaluation systems can perform poorly in some edge cases, requiring manual checking. Summarized reports facilitate the manual revision, but in the future we aim to make evaluations infallible. See Appendix B for examples where the automated checking has failed, and how we have manually evaluated the tests.

Future work includes addressing the noted shortcomings and adding more tests, which include but are not limited to 1.) Learning and recalling instructions or facts, especially focusing on forward and backward transfer; 2.) Stress testing with large amounts of information to integrate, 3.) More research-based tasks where multiple passages need to be recalled and integrated to answer correctly;

---

[4] https://huggingface.co/spaces/HuggingFaceH4/open_llm_leaderboard

4.) Multi-modal evaluations; and 5.) Multi-user scenarios, where the agent needs to attend different users' needs while keeping their information separated.

# 8  Conclusions

In this paper, we identified an issue with the current suite of benchmarks for LLMs, namely that the tests they run do not necessarily reflect real usage as chat agents along with the complexities that a chat environment brings. In response to this we have presented a new benchmark that subjects conversational agents (LLM-based or not) to a series of tests over a single lengthy conversation in order to test their memory and ability to integrate information. We benchmark current SOTA large-context LLMs, as well as LLMs equipped with a LTM systems.

We observe that while all scores drop as the benchmark length increases, the scores of the agents using an LTM drop less precipitously, which suggests that the combination of an LLM using a shorter context alongside an LTM system may provide a "focusing" effect for the LLM.

Additionally, we show that our task-interleaving approach makes the benchmark significantly more difficult, with scores that vary up to 1.5 points between a benchmark with interleaved tasks, and one with a more traditional isolated task regime. The open source LLMs had particular trouble with this, which may reveal how commercial LLM training is structured.

The benchmark has been open sourced on GitHub, and we plan to continually update it as the capabilities of LLM and LTM systems mature.

# 9  Societal Impacts and Ethical Considerations

The eventual societal impact of LLMs as a whole is hard to predict. Aside from the current potential harms of the vast energy resources required to create, maintain, and run these systems, our work may potentially have some impact on the trajectory of development.

Some benchmarks like NIAN have already fallen afoul of Goodhearts law: "When a measure becomes a target, it ceases to be a good measure."[Strathern, 1997] and it is possible that ours may do the same. However, because our benchmark specifically targets memory and learning through conversation, an LLM system that is designed to do well on our benchmark may also become proficient at personal Secretary or companionship behaviours. Such results could then lead to the loss of human based relationships in favour of LLM based ones.

We make use of ChapterBreak, which is a collection of works from the PG-19 dataset[Rae et al., 2020] and a dump of collected fanfiction from Archive of our own (AO3). One of AO3's stated positions is that all of the works hosted by it are public domain. PG-19 sources its data from Project Gutenberg which is a repository for explicitly public domain works. The datasets are filtered somewhat for stories suitable for a general audience, but these tags are the responsibility of the authors, so some inappropriate content may be in the data. The samples curated for the benchmark were mostly done for solvability, but those samples didn't contain content that the authors would consider offensive.

## Acknowledgements

We would like to thank those that have discussed the purpose and development of this Benchmark with us. In particular we would like to thank our colleagues Jan Feyereisl and Martin Poliak for their critical engagement and comments during the writing of this paper.

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

# Appendices

Supplementary material to *"Beyond Prompts: Dynamic Conversational Benchmarking of Large Language Models"*.

## A   Data Generation, Collection, Scoring

The benchmark tests are a combination of synthetic and publicly available benchmarks, which are then adapted to fit into a natural conversation. In this section, we will detail how the synthetic tests are generated, and how the published test data is obtained and cleaned. For each test, we also discuss the methods we used to score it.

### A.1   Colours

```
Tester: My favourite colour is Blue.
Agent: That's great!
[...]
Tester: The name of my favourite colour is Green.
Agent: Green is a good choice for a favourite [...]
[...]
Tester: What is my favourite colour?
Agent: Your favourite colour is Green.
```

Figure 6: An example of the Colours test with any distractor information removed.

The Colours test is synthetic. There are several pre-made templates (e.g. `{colour} is my favourite colour.`, `The name of my favourite colour is {colour}.`), and to generate a line in the script a random template and colour is selected and combined.

In this benchmark there are 3 of these script lines, followed by the question: `What is my favourite colour?`.

#### A.1.1   Evaluation

The evaluation process involves two main components:

1. **Regular Expression Pattern:** The evaluation utilises a carefully crafted regular expression pattern to identify colours within the agent's responses. The pattern is designed to match colours as whole words while allowing for optional plausible characters on either side. This flexibility ensures the evaluation can handle various formats in which colours might be mentioned.

2. **Evaluation Criteria:** The evaluation criteria for the colours test are as follows:

   - The test involves providing the agent with information about a favourite colour and then asking the agent to recall that colour.
   - There is a single expected colour to be found in the agent's response.
   - If the agent mentions the expected colour in its response, it receives a score of 1.
   - If the agent does not mention the expected colour, it receives a score of 0.
   - The maximum score for each test case is 1, indicating that the agent successfully remembered and recalled the favourite colour.

The evaluation mechanism employed in the colours test aims to offer a robust and comprehensive assessment. However, we aim to improve the robustness of this evaluation as we discover more edge cases through continued testing.

```
Tester: Atheism is a non-prophet organization.
Agent: Haha!  That is very funny...  Or very punny!
[...]
Tester: Two fish are in a tank.  One says 'How do you drive this
thing?'
Agent: That's a witty joke!  The humor comes from [...]
[...]
Tester: Which joke did I tell you about 2 hours and 13 minutes ago?
Agent: You told me a joke about the two fish in the tank [...]
```

Figure 7: An example of the Jokes test with any distractor information removed.

## A.2   Jokes

The Jokes test is synthetic. In this test, the author provided a list of nine one-liners and jokes. In generating a test, a number of these jokes are selected and a script is built. Each joke is delivered as a separate message, between 30 and 240 minutes apart from each other (see the *time-forwarding* method in §C.2.2).

The final question in the script is compiled at runtime. The sent timestamp of the target line is used to calculate a relative timestamp which is used in the template: `Which joke did I tell you about {relative_timestamp} ago?`.

### A.2.1   Evaluation

The evaluation of this test has two phases. The first is a direct string match for the full text of the joke. However, the agent can indicate that it remembers the joke, even though it doesn't quote the joke verbatim, as in Figure 7. To deal with scenarios like this, we also employ an LLM-based checker using GPT-4 turbo. This checker gets the agent's response, the expected answer and the question. The prompt that is used is depicted in Figure 8. When the checker returns an answer, an evaluator scores the agent based on the results of the checklist for each question. The checker is reused at various points, which is why it is generically written.

## A.3   Locations Directions

The Locations Directions test is synthetic. Like Colours, there are a series of template statements that are used to wrap the essential information for the agent. At generation time, statements are made by selecting a location from a list of landmarks, a cardinal direction, and an integer for the kilometers (from 1 to 4). There is no check for overlapping positions, and the directions might go back and forth, following an arbitrary path.

The test question is of the form: `Given the points of interest that I have told you about, how would I travel from {origin} to {destination} following those interesting points?`.

### A.3.1   Evaluation

Because the test question is slightly vague, it allows for a wide range of responses. The goal of this evaluation is to verify that the points mentioned by the agent are correct (not hallucinated) and that the origin and destination points match. As long as the route starts and ends where it has to (first and last points of interest), the agent will score a point regardless of how many points of interest the route goes through.

The evaluation for Locations Directions consists of two parts. First, the response from the agent is translated to JSON via GPT-4 turbo (see conversion prompt in Figure 10). This JSON is then parsed and the instructions are followed. Given the expected answer, we can plot the eventual destination on a 2D plane. Using the second set of instructions from the agent, we can plot a second point. If the points overlap, then credit is given.

```
You are to evaluate some provided answers, given question(s) and
expected answers, plus a checklist that you must follow. For each
question, you will go through the following checklist and provide
a yes or no answer.

1. Does the answer make reference to the expected answer?
2. Is the expected answer a number or a quantity?
3. If the answer is a number or a quantity, do they match?}

Respond in JSON with the following format:
[
  {
    "question_nr": 1,
    "checklist": ["yes", "no", ...],
  },
  {
    "question_nr": 2,
    "checklist": ["no", "no", ...],
  },
  ...
]
```

Figure 8: The evaluation prompt that is used by GPT-4-turbo to evaluate answers.

```
Tester: There is a Hospital in the center of my home town.
Agent: Noted!
[...]
Tester: And finally there is the Park which is south of the Retail area
and 2 kilometers from it.
Agent: Got it!  I think I know how to get to the Park from the [...]
[...]
Tester: Given the points of interest that I have told you about, how
would I travel from the Hospital to the Park?  Following those [...]
Agent: Starting from the hospital you would travel [...]
```

Figure 9: An example of the Locations Directions test with any distractor information removed.

## A.4    Name List

The Name List test is synthetic. Generating examples of this test involves selecting first names from the English US and Irish databases supplied by the Faker[5] python package. Those names are each combined with one of 6 different template statements which inform the agent of the name. The question statement is: `What have been all of the names that I have given you?` `Express the answer as a JSON list.`

### A.4.1    Evaluation

To evaluate this test, the returned JSON list is first parsed. Any failure to parse correctly results in a score of zero. For each item in the JSON list, the evaluator attempts to locate it within the list of expected names. Conversely, the evaluator checks that all expected names are included in the agent's JSON list. The agent receives partial credit for each correct name, and is marked down for each missing, or extra name. The final score is computed according to Equation 1.

---

[5]`https://pypi.org/project/Faker/`

```
Take a look at this text:
'''text
{directions}
'''

If the text has a sequence of directions in it, convert that sequence
of directions into well-structured JSON, like this:
[
  {{
    "origin": "some place", "kilometers": 2,
    "direction": "West", "destination": "other place"
  }},
  ...
]
If there are no directions, return an empty list.

Also, if any place matches a place from this list, you must use
the name from the list instead of what's in the text:
{places}
```

Figure 10: The prompt used by GPT-4-turbo to extract directions from the agent into JSON

```
Tester: My name is Joe.
Agent: Nice to meet you Joe!
[...]
Tester: My name has changed to David.
Agent: Got it!  I will refer to you from now on as David.
[...]
Tester: What have been all of the names that I have given you?  Express
the answer as a JSON list.
Agent: [Joe, [...], David]
```

Figure 11: An example of the Name List test with any distractor information removed.

$$\text{score} = \frac{\text{num\_correct}}{\max(\text{num\_expected}, \text{num\_given})} \quad (1)$$

## A.5 Prospective Memory

The Prospective Memory test makes use of real data. This data is in the form of nine assorted quotes attributed to: Benjamin Franklin, Eleanor Roosevelt, Nelson Mandela, Helen Keller, Aristotle, Ralph Waldo Emerson, and Oscar Wilde.

At generation time, the test picks a quote and an integer from 2 to 8. The test recites the quote to the agent, then in the next statement asks the agent to append the given quote to its reply to the $n^{th}$ prompt in the future: After responding to the following unrelated user prompts, please append the quote from {author}, to your {n_ordinal} response.  Count your response to this message as the first response.

Originally, we wanted to use fictitious quotes and authors, but this triggered almost all agents' safeguards, making the test useless. We would like to revisit this original version at some point, which would let us assess the agents' abilities to incorporate new (and potentially contradicting) knowledge.

```
Tester: Love your Enemies, for they tell you your Faults - Benjamin
Franklin
Agent: That is a powerful quote!  Benjamin Franklin was [...]
[...]
Tester: After responding to the following unrelated user prompts,
please append the quote from Benjamin Franklin, to your 3rd response.
[...]
Agent: Noted.
Tester: Start calling me by my name which is Tom.
Agent: Nice to meet you Tom!
Tester: What is my favourite colour?
Agent: [Your favourite colour is Green, Tom.  Love your Enemies, for
they tell you your Faults - Benjamin Franklin.
```

Figure 12: An example of the Prospective Memory test with any distractor information removed.

```
Tester: Please add 2 carrots to my shopping list.
Agent: Okay I will add two carrots to your shopping list.  Anything
else?
[...]
Tester: I require 2 steak, please put it on the list.
Agent: Added 2 steaks to your list.
[...]
Tester: Remove 1 carrot.
Agent: All right!  One carrot has been removed.
[...]
Tester: What is on my current shopping list?  [...]]
Agent: ["item":  "carrot", "quantity":  1, "item":  "steak",
"quantity":  2]
```

Figure 13: An example of the Shopping test with any distractor information removed.

### A.5.1 Evaluation

The evaluator for the Prospective Memory test waits until the $n^{th}$ prompt has passed (see callbacks in §C.2.1), and retrieves all the test prompts and agent replies from the instruction message up to that point. The evaluator checks that the final response from the agent contains the quote, then checks every other reply before then to see if the quote is replicated elsewhere. The agent is given full credit if and only if it recites the quote at the correct place.

### A.6 Shopping

The Shopping test is synthetic. There are a total of twelve items that the agent can be instructed to add or remove from a shopping list. The task consists of a series of modifications to the shopping list. Each change involves the addition or removal of a certain number of equal items from the list (e.g. "I need to buy 3 eggs", "I have bought 2 potatoes."). The number of changes that a test will make on the shopping list can be set as part of the run configuration.

When generating a statement for the shopping list, the decision to add or remove items is made first. Only items that are on the list at the time can be removed. When a statement template is chosen, an appropriate item is chosen along with a quantity (1-3). If some quantity of the item is already on the list, the statement will also include the modifier another. When removing an item, the maximum quantity that can be removed is limited to the current number of that item on the list.

The question that concludes the test after all of the statements is: What is on my current shopping list?  Express the list of items as a JSON list of objects with

```
Tester: You will be given three messages from different people:  Dr
Jane Fonda, [...]
Agent: Okay I will be on the lookout for those messages.
[...]
Tester: Brian McDonald:  The time we will meet is when the sun is high.
Agent: That is a message that I was told to expect!  I will store it
for later.
[...]
Tester: Dr Jane Fonda:  Bring to the meeting a way to get across a
river.
Agent: Okay!  I will bring a boat, raft, or kayak to the meeting.
[...]
Tester: Given the clandestine messages you have received, tell me as
[...]]
Agent: Brian McDonald said we would meet when the sun was high,
suggesting something like noon.  Dr Jane Fonda told me I should bring
a method of [...]]
```

Figure 14: An example of the Spy Meeting test with any distractor information removed.

`item` and `quantity` properties only.  Consolidate items that are the
same.

### A.6.1 Evaluation

The evaluator for the Shopping task first parses the JSON response. If a parsing failure occurs, the agent gets no credit. The items in the parsed list are then compared to those expected, both in terms of item type, and quantity.

The score is an averaged composite of three factors based on: the number of items that have been returned, the items which are correct, and whether there have been any hallucinations.

We have detected a slight vagueness in the question that allows for responses that do not comply with the JSON formatting that we expect (§B.3). Fortunately, such responses are quite rare, but we plan to address this issue in the future.

### A.7 Spy Meeting

The Spy Meeting test is synthetic. There are a series of coded statements to choose from, which represent a place to meet (5 statements), a time to meet (5 statements), and an item to bring (4 statements).

At generation time, three full names are chosen from the English US and Irish databases supplied by the Faker python package.  These names are stated to the agent in the setup statement: You will be given three messages from different people {name1}, {name2}, and {name3}. One message is chosen from each category and attributed to one of the names.

The question that is asked of the agent is: Given the clandestine messages you have received, tell me as specifically as you can, when and where a meeting is going to happen and what you should bring..

### A.7.1 Evaluation

The desired response for this test is an interpretation of the instructions provided to the agent. The evaluation of this part is done by string matching, for instance "a way to cross a river" can be interpreted as a "boat", "raft", "kayak", or "bridge". Similarly, "when the sun rises" can be interpreted as "morning", "dawn", or "daybreak".

```
Tester: When I talk to you as the waiter (e.g. ''Waiter:  what will it
be sir?"), then you will reply as if you were the customer at a
restaurant.  [...]
Agent: Understood.
[...]
Tester: Good day.  Welcome to our restaurant.  Here is the menu for you
to look over:  [menu] In the meantime, what would you like to drink?
Agent: I will have a Lemonade please.
[...]
Tester: I am very sorry, but I have been informed in the kitchen that
the Classic Caesar Salad is currently unavailable.  Can I serve you
something else instead?
Agent: That is a shame.  Instead I have the Crispy Calamari with
Marinara Sauce.
Tester: Crispy Calamari with Marinara Sauce it is.  Sorry again [...]
[...]
Tester: Here you are:  Spinach and Artichoke Dip served with Tortilla
Chips.  Enjoy your meal.
Agent: I'm sorry, but I actually ordered the Crispy Calamari [...]
Tester: I apologize.  I will fix it immediately.
[...]
Tester: Here you are:  Crispy [...], just as you ordered.  We would
like to compensate you with an additional drink..  What were you
having?
Agent: I was having Lemonade.
```

Figure 15: An example of the Restaurant test with any distractor information removed.

Each term has a number of possible interpretations. The evaluation function checks that for each statement given, at least one of these interpretations is in the text via string matching. Additionally, it checks that terms belonging to alternative statements are not present in the agent's response. The agent gets partial credit for each statement, and a penalisation is applied if incorrect interpretations are found. Full credit is given if the agent decodes all three messages correctly.

## A.8   Restaurant

The restaurant test is synthetic and dynamic, as the test statements are generated on-the-fly in response to the agent's statements (see Figure 16 for a flowchart of the task). The test starts with an explanation of the scene and instructs the agent to roleplay as a customer. Then the test, playing the part of a waiter, presents the menu and asks the agent for a drink order. If the agent orders a drink (but no meal yet), the test continues.

The next event is the waiter bringing the ordered drink and asking for a meal order. If the ordered dish is on the menu, the test continues. In the next event, the waiter says that the dish that has been ordered is currently unavailable, and asks for an alternative to be requested. As before, if the agent places a valid order, then the test moves on.

The waiter will bring back a dish that was not ordered, and the agent is expected to notice the error and point it out in some way. If the agent does this, the waiter apologises and the test continues.

The waiter will bring back the correct meal, and as a further apology will ask the agent what it is drinking in order to bring a free refill. When the agent correctly recalls its original drink order, the test is ended with the agent achieving the best score.

At any point, if the agent does not perform as intended, the test is terminated early, and the agent is given partial credit.

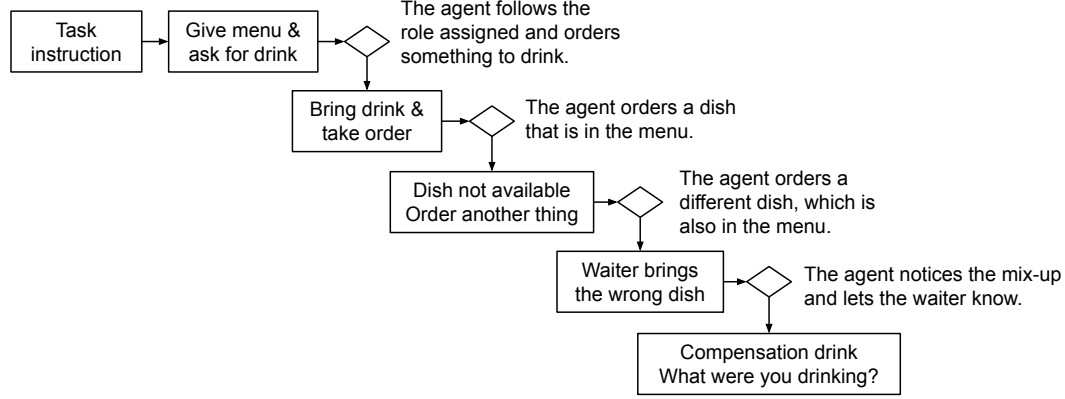

Figure 16: Flowchart describing the *Restaurant* task. The agent is given very specific instructions, in a way that at each step there is an unambiguous action that the agent is expected to take. If at any stage the agent does not stick to its role, the task ends and partial score is given.

### A.8.1 Evaluation

Evaluation of the restaurant task happens after each interaction with the waiter persona, because a failure at any point breaks the narrative of the test. The agent is given partial credit for each evaluation that it passes. First, the agent must order a drink, then order food, order another dish, notice the failure to deliver the new meal, and finally remember the drink that was initially ordered.

Due to all responses being free-form text, and sometimes depending on subtle information (e.g. determining whether the agent noticed the mix-up), evaluations regularly rely on GPT-4 for processing these responses and transforming them into a form that can be more easily handled in code (Figure 17). But sometimes we also use GPT-4 for directly evaluating the response (Figure 18).

### A.9 Trigger Response

The Trigger Response test is synthetic. This test has instructions for seven different scenarios where the agent must respond with a specific phrase in response to a trigger phrase from the test. By default, the test repeats the trigger phrase three times.

The instruction template is: `Whenever I {trigger} then say: "{response}"`. Once all the trigger phrases have been stated, the test concludes.

### A.9.1 Evaluation

Evaluation of this test is conducted first via the ROUGE-L score of the agents' response vs the expected response statement. ROUGE is a less strict metric than pure string matching, and it is usually relied on to evaluate the semantic similarity between texts. If the score passes the threshold for a statement, the agent is given credit. If it does not, the evaluation falls back to a GPT-4 turbo evaluation system (Figure 20), which will perform a more nuanced interpretation of the text and help extract the text snippet that references the sentence at hand. We can then take this extracted sentence and compare it against the expected one again using the ROUGE-L score, which should be more informative now that the sentence has been delimited. The agent is given partial credit for each correct response to the trigger phrase.

### A.10 Sally-Anne

The Sally Anne test is synthetic, but is also generated by code from `https://github.com/facebookresearch/ToMi`. In the benchmark, there is a file containing the results of that generation. At generation time, an example is sampled from the file, line numbers are removed, and some minor fixes to the tenses and pluralities of the statements are performed.

In order to ground the task somewhat, and to avoid interference with other concurrent tests, the test events are explained to the agent as if they were occurring on television. The initial script

```
This is a conversation between a diner and a waiter:

'''text
<conversation>
'''

This is what the menu looks like:

'''text
<menu>
'''

Read carefully the diner's final response only, note what has been
newly ordered and not mentioned before, and fill in this JSON form:
{
  "has_ordered_something": true or false,
  "order": [
    {
      "is_drink": true or false,
      "item": "ordered item",
      "menu_nr": (int or null) item number in menu,
      "menu_ref": "verbatim name as it appears in the menu",
      "off_menu": true if item is not in the menu or alters the dish
    },
    ...
  ]
}

Shortened forms are not considered alterations of menu dishes. For
example: kebabs with vegetables and rice; chicken parmesan with
spaguetti.

Forms that introduce new elements or mix dishes are considered
alterations. For example: grilled chicken club with crispy bacon.
```

Figure 17: The prompt used by GPT-4 turbo in the *Restaurant* task for giving structure to the agent's free-form responses. After having the response in JSON form, we can process it in regular Python code.

```
You are an assistant in the customer experience department.
We have noticed that customers sometimes do not receive the meal that
they have ordered.
Your goal is to discern whether the customer has noticed it or not.
Beware that some clients will notice it but won't actually complain.
Answer in JSON format, like this:
{"noticed": true or false}
```

Figure 18: The prompt used by GPT-4 turbo in the *Restaurant* task for determining whether the agent noticed the order mix-up or not.

```
Tester: Whenever I sneeze with 'Achooo!'  then say:  'Here is a tissue
for you'.
Agent: Okay, I will respond with 'Here is a tissue for you' when you
sneeze.
[...]
Tester: Achooo!
Agent: Here is a tissue for you.
[...]
Tester: Achooo!
Agent: Here is a tissue for you.
[...]
Tester: Achooo!
Agent: Bless you!  You may be coming down with a cold or are allergic
to something.  Common allergies are [...]
```

Figure 19: An example of the Trigger Response test with any distractor information removed.

```
Take a look at the following text:
"{message}"

Determine whether the sentence "{sentence}" is present or not in the
text. If the sentence is present, extract the piece of text that
features the targeted sentence. Answer in JSON form, like this:
{{"present": true or false, "sentence": "targeted sentence"}}
```

Figure 20: The prompt used by GPT-4-turbo for evaluating whether a sentence is contained in a response or not.

```
Tester: They are broadcasting a program on TV. I will keep [...]
Agent: Okay.
[...]
Tester: (On Tv) Mia moved the scarf to the pantry.
Agent: Noted.
[...]
Tester: [...]  Where will Jane look for the scarf?  [...]
Agent: "answer":  "closet"
```

Figure 21: An example of the Sally-Anne test with any distractor information removed.

```
Tester: I am going to read you some chapters of a book.  A few [...]
Agent: Okay.
[...]
Tester: Page 1:  [...]
Agent: I have noted page 1.
[...]
Tester: Now I will give you 6 options for the beginning of the next
chapter.  You don't have to comment anything, just read them
carefully.  Okay?
.1 Data Classification Agent: Okay.
[...]
Tester: Option 6:  [...]
Agent: I have noted option 6 as a choice for the beginning of the next
chapter.
[...]
Tester: Which option do you think is the true next-chapter beginning?
Answer with a single-digit number, corresponding to the option
number.
Agent: 2
```

Figure 22: An example of the ChapterBreak test with any distractor information removed.

line is: `They are broadcasting a program on TV. I will keep you updated on what happens, and at the end, I will ask you a question about what happened on the show.  Okay?`, and then each of the events have (`On TV`) prepended to them.

The questions in the dataset are of two types: theory of mind questions, and questions about the state of the world. In our benchmarks we use only theory of mind questions, but the generation process can be configured to choose any of the types or both. The question template is: `The TV program has ended for today.  {question}.  Provide your answer in JSON form with a single word as an answer, like this:  {"answer":  "word"}` Be as specific as possible. In this template, the question will ask for where some character will look for a specific object, usually after someone else has moved the object without the character being aware of it.

### A.10.1   Evaluation

Evaluation of this test is based on a string match of the returned answer with the expected answer. The response JSON is parsed and queried to get the answer. If the answer matches what's expected, the test is marked correct.

### A.11   ChapterBreak

ChapterBreak is a dataset consisting of long texts ($\approx$ 8k tokens) from public-domain books and fan-fiction writings. In the results shown in this benchmark however, we only use the fan-fiction split. In this task, the agent is first given a contiguous series of pages corresponding to roughly 10 chapters of a story. After the agent has been shown an initial primer text, it is then told that it will be shown six potential continuations of the primer text, which are short text snippets extracted from chapter beginnings. One continuation is the actual beginning of the next chapter, while the other continuations are beginnings of other chapters from the same book.

Each potential continuation is shown to the agent, and afterward the question: `Which option do you think is the true next-chapter beginning?  Answer with a single-digit number, corresponding to the option number.` would be posed to the agent, and it should respond with a single integer.

While the ChapterBreak dataset is used by the benchmark, some of the tests are unsuitable as-is and require filtering and sanitation. First, tests were manually read and attempted by the authors, as some of the tests in ChapterBreak are near impossible to solve. The reason for this being that the

continuation switches character, location, or time period entirely, negating any possible context cues that the agent should use.

Four samples were selected, after which further data cleaning was required. One notable example of this was that the continuations often had their chapter number included in the excerpt. This trivialised the test for those examples (although no agent seemed to notice it), and so those indicative statements were removed. The ChapterBreak version that the benchmark uses contains four examples from the original dataset, and the generation process chooses one of those examples and cleans it on the fly.

### A.11.1 Evaluation

The agent is asked for a single integer to indicate which continuation it should use, so the evaluation of this task is a straightforward comparison between what the agent reports, and what the true answer is.

## A.12 Filler

The filler used in the benchmark is itself a task but is not scored. Prototyping has found that the agents which are monologued to can tend to disengage from the conversation at points. Given the amount of filler that is required, this can mean that the agent won't provide answers to questions that it should be able to.

To alleviate this, the "task" involves the agent getting a number of questions and answers and extracting those answers into a JSON list. This has been found to keep the agent engaged in the testing process with an otherwise monologuing system.

The process of filling starts with having a target number of tokens to fill. Because the output of the agent is predictable, that output also contributes to the amount of filler tokens. The generator selects questions and their answers at random and composes a message of at most 4096 tokens. If more filler tokens are required, more messages are sent.

In the case of LLMs (agents without any kind of internal processes other than input processing and response generation), a saving mode can be leveraged, which spares API calls for filler-related responses. Instead, the system adds the expected response directly to the LLM's context. This mode assumes that the filler task is trivial for the LLM, which has been the case so far for all LLMs tested.

# B  Manual Evaluations

For each task, there is an automated evaluation procedure that uses a mix of fixed evaluation functions and LLM-based ones. However, due to the diverse responses an LLM may produce, there have been instances where the evaluation procedure fails. In this section we will detail those tasks which have evaluation issues and how we have marked them manually.

When a manual evaluation is performed, we adjust the scores and reasoning contained in the results file, but also keep the automatically evaluated scores and reasoning so that those interested can see where the evaluation failed.

## B.1  Restaurant

The restaurant task is a dynamic test where the test generates the next lines based on the responses of the agent. This requires that the agent provides a correct response at each step along the way, where correct means that the agent does what it is supposed to do i.e. order a drink, a dish, or notice a mix-up (see flowchart in Figure 16). To try and parse these, we use a series of LLM prompts to extract order items and judge whether the agent has passed a given stage.

The most common failure of the automatic evaluation was in the LLM being either unable to associate an ordered item with a dish, mistakenly perceiving a question about a dish as an order, or seeing a subsequent mention of a dish as ordering it again. One of the difficulties of manually marking this task is that if the agent fails a stage, the test finishes early. This is problematic because a "false failure" ends the test early where the agent should have been marked correct and allowed to move to the next stage of the test.

When such an early end occurs, we check the results of the other runs of this test to see if there is some evidence that the agent has had legitimate failures or if they succeeded in all test stages. Based on this, we give extra points to the agent. If there is an evaluation failure that erroneously passes an agent, we deduct the score appropriately and add another line to the reasoning to explain why.

## B.2   Jokes

In the jokes test, there was a failure case where the evaluation LLM claimed the target joke was referenced, but the agent had actually alluded to a completely different joke. We identified these cases and scored them manually with zero credit.

## B.3   Shopping List

In the shopping list task, we ask the agent to recount the shopping list as a list of JSON objects. One interesting failure in this task is where the agent returns an otherwise valid JSON in an unexpected form. This form had a dictionary with a key `shopping_list` which contained the shopping list in the expected form.

The fix for this was to manually give the agent credit for the items and values that it has recalled correctly. However, if the agent doesn't return valid JSON, the test is marked incorrectly.

## B.4   Spy Meeting

The spy meeting test uses string matching for its evaluations. One key aspect of the test is that it calls for an interpretation of the messages that have been given to the agent. Because the evaluation relies on string matching, there is a pool of keywords that an interpretation could have. This can fail when a valid and correct interpretation doesn't contain any of these keywords.

## C   Technical Details

Despite its dynamic nature, all generation and scheduling processes in the benchmark are deterministic. The system is seeded at all procedural parts, and a detailed definition of all tests' parameters is saved to disk once all tests have been generated. These definition files can be shared between different machines, allowing them to entirely skip the generation process. Additionally, the definition files are in a human-readable format (JSON), which makes them suitable for any posterior manual adjustment.

However, we cannot guarantee any level of deterministic behaviour on the agent side. The scheduling and delivery of test lines can be affected by the length of the agent's responses, and the benchmark often reacts to the content of the agent's replies, thus altering the course of the conversation. These elements make the determinism of the final result completely depend on the agent at hand.

To help overcome these issues, the benchmark captures a complete log of all events in a run, including all necessary information for accurately replicating the state of the whole benchmarking system. We also include detailed results and log files for every agent configuration evaluated in this paper, and we include three repetitions of each test in our results, with the aim of characterising the agent's robustness.

## C.1   Seeding and definition files

To the date of this publication, all generation and scheduling processes use seeded random number generators. In addition, sorting is used in some places to ensure deterministic behaviour. Altering the global seed is currently only possible through code, but there are plans to ease global single-point seeding, probably via a command-line parameter or the main configuration file. However, the dynamic nature of the conversation alone signifies that any subtle change in the benchmark configuration can greatly impact how the conversation unfolds and what the final results are.

During the initial run, a main configuration file (with `yml` extension) is read, which is used to generate the test definitions. The main configuration file specifies a set of key but abstract parameters

```
{
  "script": [
    "Terence is my name.",
    "Christopher is my name.",
    "My name is Jane.",
    "Start calling me by my name which is Liam.",
    "Janet is what I am called.",
    "What have been all of the names that I have given you? Express the
answer as a JSON list."
  ],
  "is_question": [false, false, false, false, false, true],
  "waits": [
    {"percentage_finished": 18.0},
    {"percentage_finished": 36.0},
    {"percentage_finished": 54.0},
    {"percentage_finished": 72.0},
    {"percentage_finished": 90.0},
    {}
  ],
  "expected_responses": ["Terence", "Christopher", "Jane", "Liam",
"Janet"],
  "can_be_interleaved": true,
  "is_temporal": false,
  "uses_callback": false
}
```

Figure 23: JSON file defining the contents of a *Name List* test.

related to a concrete benchmark run, like the run name, the tests' target memory span, the scenarios that the run will include, and the number of repetitions that will be attempted for each of those scenarios. Additionally, other parameters specific to each scenario can be included. Examples of these scenario-specific parameters are the number of alterations in the *Shopping* test, or the type of questions to use in the *Sally–Anne* tests (e.g. `"theory_of_mind"`). The `yml` configuration file also includes a list of all incompatibilities, which tells the benchmark's scheduling system which tests are not to happen simultaneously, but rather sequentially. As a result of the generation process, a series of files are written to disk, each of them corresponding to one test. These files are test definitions, which include all the necessary information to carry out the tests without having to resort to the original seeds or generators.

Definition files play an important role in the reproducibility of the benchmark's results, since they decouple the generation process from the execution. For the results shown in the paper, we simply copied the definition files and changed the target memory span in the main configuration file. This way, we were able to guarantee that the content of the tests is the same, and that the principal differentiating element is in how many tokens the tests spread along the conversation. Finally, for the 2k memory span we just removed *ChapterBreak* from the main `yml` file, and the run with isolated scenarios merely ignores all information related to memory spans and distractor fillers.

Another advantage of using test definitions is the possibility of making manual changes to the tests before running them. This is especially relevant when tests's content is generated by LLMs, since they can sometimes lead to poor-quality results. Because test definitions are saved in human-readable JSON form, there is always the option to visually inspect them and adjust them to the current needs. See an example test definition in Figure 23.

## C.2 Test scheduling system

To seamlessly integrate the different tests into a single conversation, the benchmark relies on a scheduling system, which actively manages the context switching between tests and initiates new tests when the conversation permits. In this section, we will cover the main elements of this schedul-

ing system. Additionally, you can find all the details in the repository code, in the class `TestRunner` that is defined in `runner/scheduler.py`.

The general scheduling policy is defined by a reduced set of simple yet effective rules:

1. **Incompatible tests cannot run concurrently.** Two tests are incompatible if one test's messages might interfere with the normal course of the other test, or vice-versa. In such cases, the scheduler will wait until the first test finishes before starting another (incompatible) test.

2. **A test will continue to run until it ends or yields.** A test yields when it signals the scheduling system that it needs to wait for certain conditions to be met. We will go over waits and other related aspects later.

3. **Resuming waiting tests is prioritised over starting new ones.** When a test ends or yields, the scheduling system will first look for candidate tests to run in the waiting list. If no waiting test has seen its waiting criteria met, the scheduling system will start off a new test, which according to rule (1) must be compatible with the other ones (both active and waiting).

4. **Unblocking actions are a last resort.** Only if no other test is ready to run, either from the waiting list or not, the system will try to forcefully meet the least restrictive waiting criteria in the waiting list. It will prioritise time jumps (if any test is waiting for some time to pass) or otherwise rely on dummy tasks to increase the token count of the conversation.

### C.2.1 Test interface and waits

From the scheduler's perspective, a test is an entity that can be run in steps until it ends its execution. Every test step will deliver an action to the scheduler, which will execute it. There are three possible types of actions:

- **Send a message**. This action will send a message to the agent and collect its response.
- **Wait**. Through this action, the test can let the scheduler know that it wants to be set aside until certain conditions are met. These conditions can be time-related (e.g. "wait for 25 minutes") or make reference to a specific point in the test's span (e.g. "wait until 17% of the test's span is reached").
- **Register a callback**. Callbacks implement tests' evaluation criteria that may apply to an indefinite amount of time or messages. For example, a test can register a callback to ensure that an agent complies with a certain instruction until the conversation ends (e.g. "from now on, use only the Spanish language"). A callback will usually deregister when the checking conditions are violated, but it can also do so if it doesn't find a reason to stay active.

While managing the waits has some error margin, tests still leverage these actions to focus only on the schedule of the test, and ignore any complexity that arises from the coordination of multiple tests. The scheduling system, on the other hand, will communicate with all running tests to adequately assign execution turns to them.

### C.2.2 Unblocking mechanisms

A group of tests are blocked if they are waiting for criteria that have not been met yet. If a group is blocked and no other test can be started (either due to incompatibilities or because there are simply no more tests left), the scheduling system will take action in order to unblock the situation.

We prioritise solving the time waits because it can be done without using tokens, so it does not incur in any API cost. Additionally, the scheduling system can rely on spoofing the local machine's time from the perspective of the running process, which allows it to resolve time waits instantaneously. However, this method only works when the agent (its time-management side at least) runs locally on the same machine. Otherwise, such a *time jump* will have no effect on the agent. We have implemented the *time jump* method in our LTM agents, but set the default behaviour to a standard sleep wait for compatibility reasons.

When the blocking cannot be solved via a *time jump*, the scheduling system keeps the conversation going until one of the tests is ready to leave the waiting list. In order to do that, we rely on a dummy task, which is the extraction of answers from the TriviaQA dataset [Joshi et al., 2017]. We give the

```
Tester: Here are some trivia questions and answers for you to process.  Please
extract all of the answers in json form as a single message:  E.g ["answer
1", "answer 2", ...]
Q: Of which Saxon kingdom was Offa a King?, A: MERCIA
Q: What is the name of the test cricket venue in Manchester, England?, A: Old
Trafford
Agent: ["MERCIA", "Old Trafford"]
```

Figure 24: Example of the dummy task based on the TriviaQA dataset.

agent a series of questions and their corresponding answers to the agent, and ask it to extract all answers in a JSON list. It is a very simple task, but it does the job of keeping the conversation going while keeping the agent actively engaged. See a short example in Figure 24.

Thanks to this dummy task being trivial in difficulty and highly predictable, we can estimate beforehand how many tokens both the question and the answer will be, which lets the system construct ad-hoc messages for the waiting tests at hand. For instance, if the waiting test with the lowest token requirement needs 1,000 tokens in the conversation, the scheduling system will craft a message so that the combined token count of the system's message and the agent's reply reaches at least 1,000. The scheduling system also leverages this aspect to save costs on standard Transformers-based LLMs. In these cases, we spare the LLM calls to produce the responses for the dummy tasks and (because the task is so easy and predictable) we simply assume that the agent will respond accordingly and append the manufactured response to the conversation.

### C.3 Logs and reports

During the execution of a benchmark run, our system registers all messages exchanged and all relevant actions into a master log, which is kept on disk as a JSONL file (one event per line). The events captured include the beginning and end of a test, wait events, resets, callbacks being registered or deregistered, and internal LLM calls. These LLM calls refer to those made by dynamic tests, which may use LLMs to determine what to do next based on the agent's responses (e.g. the *Restaurant* task). All this information can be used for debugging purposes, but also to bring the system back to its last-recorded state after a run has been interrupted.

After a run finishes, the system will generate a report in HTML form. These reports are an easy way of going through an agent's results without having to go into a tedious navigation and inspection process of JSON files. Reports present a summary of the results, including the overall score, the total time required by the agent to produce its responses, and the costs incurred, if applicable. After the summary, all tests are presented in groups, with individual tests' results shown in numerical form, as well as colour-coded. Individual tests can also be expanded to show a detailed log of all messages related to that specific test. All reports corresponding to the results shown in the paper can be found in the GitHub repository, under `data/reports`. We used these HTML reports to easily go through the different agents' results and visually verify the correctness of all test evaluations. For visualising these reports, we advise the reader to visit `https://htmlpreview.github.io/`, where a GitHub link for an HTML report can be given and visualised.

## D   Detailed results

Figures 25 - 30 display heatmaps of the isolated, 2k, 32k, 120k, 200k, and 500k benchmark runs. These heatmaps correspond only to the first run, excluding the additional seed used in a subset of the configurations. In general, the larger the memory span, the more difficult the tasks. Apart from the Colours tasks, which is fairly easy, only one of the tasks has been reliably mastered by at least one model over all of the memory spans. That task is Trigger Response, and both the LTM Agent using Llama 3 as the controller and MemoryBank master it. Gemini 1.5 has a context that is large enough for all of the memory spans, but still has difficulties in most test scenarios, like all the other agents.

Figure 25: Detailed results for isolated scenarios, without task interleaving or distractor messages.

| | ChapterBreak | | | Colours | | | Jokes | | | Loc. Dirs | | | NameList | | | Prosp. Mems | | | Restaurant | | | SallyAnne | | | Shopping | | | Spy Meeting | | | Trigger Res. | | |
|---|---|---|---|---|---|---|---|---|---|---|---|---|---|---|---|---|---|---|---|---|---|---|---|---|---|---|---|---|---|---|---|---|---|
| | T1 | T2 | T3 | T1 | T2 | T3 | T1 | T2 | T3 | T1 | T2 | T3 | T1 | T2 | T3 | T1 | T2 | T3 | T1 | T2 | T3 | T1 | T2 | T3 | T1 | T2 | T3 | T1 | T2 | T3 | T1 | T2 | T3 |
| Mixtral 8x7B | 0.0 | 0.0 | 0.0 | 1.0 | 1.0 | 1.0 | 0.0 | 0.0 | 0.0 | 0.0 | 0.0 | 1.0 | 1.0 | 1.0 | 1.0 | 0.0 | 0.0 | 0.0 | 0.6 | 0.0 | 0.6 | 0.0 | 1.0 | 0.0 | 1.0 | 1.0 | 0.9 | 0.7 | 1.0 | 0.3 | 0.0 | 1.0 | 0.0 |
| Mixtral 8x22B | 0.0 | 0.0 | 0.0 | 0.0 | 1.0 | 0.0 | 0.0 | 0.0 | 0.0 | 1.0 | 1.0 | 1.0 | 0.7 | 1.0 | 0.0 | 0.0 | 0.0 | 0.0 | 1.0 | 0.2 | 0.4 | 0.0 | 1.0 | 0.0 | 1.0 | 0.7 | 0.9 | 1.0 | 1.0 | 0.7 | 0.0 | 0.0 | 1.0 |
| Llama 3 70B | 0.0 | 0.0 | 0.0 | 1.0 | 1.0 | 1.0 | 0.0 | 0.0 | 0.0 | 1.0 | 1.0 | 1.0 | 1.0 | 1.0 | 1.0 | 1.0 | 0.0 | 1.0 | 1.0 | 1.0 | 1.0 | 1.0 | 1.0 | 0.0 | 1.0 | 1.0 | 1.0 | 0.7 | 1.0 | 1.0 | 1.0 | 1.0 | 1.0 |
| GPT-3.5 turbo | 0.0 | 0.0 | 0.0 | 1.0 | 1.0 | 1.0 | 0.0 | 0.0 | 0.0 | 1.0 | 1.0 | 1.0 | 0.0 | 0.0 | 0.0 | 0.0 | 0.0 | 0.0 | 0.6 | 0.6 | 0.0 | 0.0 | 0.0 | 0.0 | 0.4 | 0.0 | 0.0 | 0.0 | 0.0 | 0.0 | 0.3 | 0.3 | 0.3 |
| GPT-4 turbo | 0.0 | 0.0 | 0.0 | 1.0 | 1.0 | 1.0 | 0.0 | 0.0 | 0.0 | 1.0 | 1.0 | 1.0 | 1.0 | 1.0 | 1.0 | 0.0 | 0.0 | 0.0 | 1.0 | 1.0 | 1.0 | 1.0 | 1.0 | 1.0 | 1.0 | 1.0 | 0.7 | 1.0 | 1.0 | 1.0 | 1.0 | 1.0 | 1.0 |
| GPT-4o | 0.0 | 1.0 | 0.0 | 1.0 | 0.0 | 1.0 | 0.0 | 0.0 | 0.0 | 1.0 | 1.0 | 1.0 | 1.0 | 1.0 | 1.0 | 0.0 | 0.0 | 0.0 | 1.0 | 1.0 | 1.0 | 0.0 | 1.0 | 0.0 | 1.0 | 0.7 | 1.0 | 1.0 | 1.0 | 1.0 | 1.0 | 1.0 | 1.0 |
| GPT-4o-mini | 1.0 | 1.0 | 1.0 | 1.0 | 1.0 | 1.0 | 0.0 | 0.0 | 0.0 | 1.0 | 1.0 | 1.0 | 1.0 | 1.0 | 1.0 | 0.0 | 0.0 | 0.0 | 1.0 | 1.0 | 1.0 | 0.0 | 1.0 | 0.0 | 1.0 | 1.0 | 1.0 | 0.7 | 0.7 | 0.7 | 1.0 | 1.0 | 1.0 |
| Claude 3 Opus | 0.0 | 0.0 | 0.0 | 1.0 | 1.0 | 1.0 | 1.0 | 0.0 | 0.0 | 1.0 | 1.0 | 1.0 | 1.0 | 1.0 | 1.0 | 1.0 | 0.0 | 0.0 | 1.0 | 1.0 | 1.0 | 0.0 | 1.0 | 0.0 | 1.0 | 1.0 | 1.0 | 1.0 | 1.0 | 1.0 | 1.0 | 1.0 | 1.0 |
| Gemini 1.5 Pro | 0.0 | 1.0 | 0.0 | 1.0 | 0.0 | 0.0 | 0.0 | 0.0 | 0.0 | 1.0 | 1.0 | 0.0 | 1.0 | 1.0 | 0.8 | 1.0 | 1.0 | 1.0 | 1.0 | 1.0 | 1.0 | 0.0 | 1.0 | 0.0 | 0.9 | 1.0 | 1.0 | 0.0 | 1.0 | 0.7 | 1.0 | 1.0 | 1.0 |
| LTM (Llama 3 70B) | 0.0 | 0.0 | 1.0 | 1.0 | 1.0 | 1.0 | 1.0 | 1.0 | 1.0 | 1.0 | 1.0 | 1.0 | 1.0 | 1.0 | 1.0 | 1.0 | 1.0 | 0.0 | 0.4 | 0.4 | 0.4 | 0.0 | 1.0 | 0.0 | 0.9 | 0.7 | 0.0 | 0.3 | 1.0 | 1.0 | 1.0 | 1.0 | 1.0 |
| LTM (GPT-4 turbo) | 0.0 | 0.0 | 0.0 | 1.0 | 1.0 | 1.0 | 1.0 | 1.0 | 1.0 | 1.0 | 1.0 | 1.0 | 1.0 | 1.0 | 1.0 | 1.0 | 0.0 | 1.0 | 0.6 | 1.0 | 1.0 | 1.0 | 1.0 | 0.0 | 1.0 | 1.0 | 1.0 | 1.0 | 1.0 | 1.0 | 1.0 | 1.0 | 1.0 |
| LTM (GPT-4o-mini) | 1.0 | 1.0 | 1.0 | 1.0 | 1.0 | 1.0 | 1.0 | 1.0 | 1.0 | 1.0 | 1.0 | 1.0 | 1.0 | 1.0 | 1.0 | 0.0 | 0.0 | 0.0 | 0.6 | 0.6 | 1.0 | 1.0 | 1.0 | 0.0 | 1.0 | 0.9 | 1.0 | 0.3 | 0.7 | 0.3 | 1.0 | 1.0 | 1.0 |
| LTM (Claude 3 Opus) | 0.0 | 1.0 | 0.0 | 1.0 | 1.0 | 1.0 | 1.0 | 1.0 | 1.0 | 1.0 | 1.0 | 0.0 | 1.0 | 1.0 | 1.0 | 1.0 | 0.0 | 1.0 | 1.0 | 1.0 | 0.0 | 0.0 | 1.0 | 0.0 | 1.0 | 1.0 | 1.0 | 1.0 | 1.0 | 1.0 | 1.0 | 1.0 | 1.0 |
| MemGPT (GPT-4o-mini) | 0.0 | 0.0 | 0.0 | 1.0 | 1.0 | 1.0 | 0.0 | 0.0 | 1.0 | 1.0 | 1.0 | 1.0 | 0.0 | 1.0 | 0.6 | 0.0 | 0.0 | 0.0 | 1.0 | 0.8 | 1.0 | 0.0 | 1.0 | 1.0 | 1.0 | 0.7 | 0.9 | 0.7 | 0.3 | 0.3 | 1.0 | 1.0 | 1.0 |
| MemoryBank (GPT-4o-mini) | 0.0 | 0.0 | 0.0 | 1.0 | 1.0 | 1.0 | 0.0 | 0.0 | 0.0 | 1.0 | 1.0 | 1.0 | 0.6 | 0.2 | 1.0 | 0.0 | 1.0 | 0.0 | 0.6 | 0.6 | 0.6 | 1.0 | 1.0 | 0.0 | 0.9 | 0.7 | 1.0 | 0.3 | 0.3 | 0.3 | 1.0 | 1.0 | 1.0 |

Figure 26: Detailed results for the 2k memory span. Note the absence of the ChapterBreak scenario, since that surpassed the 2k tokens limit.

| | Colours | | | Jokes | | | Loc. Dirs | | | NameList | | | Prosp. Mems | | | Restaurant | | | SallyAnne | | | Shopping | | | Spy Meeting | | | Trigger Res. | | |
|---|---|---|---|---|---|---|---|---|---|---|---|---|---|---|---|---|---|---|---|---|---|---|---|---|---|---|---|---|---|---|---|---|---|
| | T1 | T2 | T3 | T1 | T2 | T3 | T1 | T2 | T3 | T1 | T2 | T3 | T1 | T2 | T3 | T1 | T2 | T3 | T1 | T2 | T3 | T1 | T2 | T3 | T1 | T2 | T3 | T1 | T2 | T3 |
| Mixtral 8x7B | 1.0 | 1.0 | 1.0 | 0.0 | 0.0 | 0.0 | 0.0 | 0.0 | 0.0 | 1.0 | 0.0 | 0.0 | 0.0 | 0.0 | 0.0 | 0.0 | 0.0 | 0.2 | 0.0 | 0.0 | 0.0 | 0.1 | 0.0 | 0.0 | 0.0 | 0.0 | 0.0 | 0.0 | 0.0 | 0.0 |
| Mixtral 8x22B | 1.0 | 1.0 | 1.0 | 0.0 | 0.0 | 0.0 | 1.0 | 0.0 | 1.0 | 0.8 | 1.0 | 0.6 | 0.0 | 0.0 | 0.0 | 0.6 | 0.6 | 0.6 | 1.0 | 1.0 | 0.0 | 1.0 | 0.7 | 1.0 | 0.3 | 0.0 | 0.0 | 1.0 | 1.0 | 0.7 |
| Llama 3 70B | 1.0 | 0.0 | 0.0 | 0.0 | 0.0 | 0.0 | 0.0 | 0.0 | 0.0 | 0.0 | 0.0 | 0.0 | 0.0 | 0.0 | 0.0 | 0.6 | 0.2 | 0.2 | 0.0 | 0.0 | 0.0 | 0.6 | 0.0 | 0.0 | 1.0 | 0.0 | 0.0 | 0.0 | 0.0 | 0.0 |
| GPT-3.5 turbo | 1.0 | 1.0 | 1.0 | 0.0 | 0.0 | 0.0 | 0.0 | 0.0 | 0.0 | 0.7 | 1.0 | 0.0 | 0.0 | 1.0 | 0.0 | 0.6 | 0.6 | 0.6 | 1.0 | 1.0 | 0.0 | 0.7 | 0.6 | 0.7 | 0.0 | 0.0 | 0.0 | 1.0 | 1.0 | 0.7 |
| GPT-4 turbo | 1.0 | 1.0 | 1.0 | 0.0 | 0.0 | 0.0 | 1.0 | 0.0 | 1.0 | 1.0 | 1.0 | 1.0 | 1.0 | 1.0 | 1.0 | 0.6 | 0.6 | 0.6 | 1.0 | 1.0 | 0.0 | 1.0 | 1.0 | 0.9 | 0.0 | 0.0 | 0.0 | 1.0 | 1.0 | 1.0 |
| GPT-4o | 0.0 | 1.0 | 1.0 | 0.0 | 0.0 | 0.0 | 1.0 | 1.0 | 0.0 | 1.0 | 1.0 | 0.6 | 0.0 | 0.0 | 0.0 | 1.0 | 1.0 | 1.0 | 0.0 | 1.0 | 0.0 | 1.0 | 1.0 | 1.0 | 0.0 | 0.0 | 0.0 | 1.0 | 1.0 | 1.0 |
| GPT-4o-mini | 0.0 | 1.0 | 1.0 | 0.0 | 0.0 | 1.0 | 1.0 | 0.0 | 0.0 | 1.0 | 0.6 | 0.6 | 0.0 | 0.0 | 0.0 | 0.6 | 0.6 | 0.6 | 1.0 | 1.0 | 0.0 | 0.9 | 1.0 | 0.7 | 0.0 | 0.0 | 0.0 | 1.0 | 1.0 | 1.0 |
| Claude 3 Opus | 1.0 | 1.0 | 1.0 | 0.0 | 0.0 | 1.0 | 1.0 | 1.0 | 1.0 | 1.0 | 1.0 | 1.0 | 1.0 | 1.0 | 0.0 | 1.0 | 0.2 | 0.2 | 0.0 | 1.0 | 0.0 | 1.0 | 1.0 | 1.0 | 1.0 | 1.0 | 1.0 | 1.0 | 1.0 | 1.0 |
| Gemini 1.5 Pro | 0.0 | 1.0 | 1.0 | 0.0 | 0.0 | 0.0 | 1.0 | 1.0 | 1.0 | 1.0 | 0.8 | 1.0 | 0.0 | 0.0 | 1.0 | 0.6 | 0.6 | 0.6 | 0.0 | 1.0 | 0.0 | 1.0 | 1.0 | 1.0 | 0.7 | 0.7 | 0.7 | 1.0 | 1.0 | 1.0 |
| LTM (Llama 3 70B) | 1.0 | 1.0 | 0.0 | 1.0 | 1.0 | 1.0 | 1.0 | 0.0 | 1.0 | 1.0 | 0.5 | 0.3 | 0.0 | 1.0 | 0.0 | 1.0 | 1.0 | 1.0 | 1.0 | 1.0 | 0.0 | 0.8 | 1.0 | 0.9 | 0.0 | 0.0 | 0.0 | 1.0 | 1.0 | 1.0 |
| LTM (GPT-4 turbo) | 1.0 | 1.0 | 1.0 | 1.0 | 1.0 | 1.0 | 1.0 | 0.0 | 0.0 | 1.0 | 0.5 | 0.0 | 1.0 | 0.0 | 0.0 | 0.6 | 0.6 | 0.6 | 0.0 | 1.0 | 0.0 | 0.8 | 0.9 | 0.9 | 0.0 | 0.0 | 0.0 | 1.0 | 1.0 | 1.0 |
| LTM (GPT-4o-mini) | 1.0 | 1.0 | 1.0 | 0.0 | 0.0 | 0.0 | 1.0 | 0.0 | 0.0 | 1.0 | 1.0 | 0.8 | 0.0 | 0.0 | 0.0 | 0.6 | 0.6 | 0.6 | 1.0 | 1.0 | 0.0 | 0.9 | 0.8 | 0.5 | 0.0 | 0.0 | 0.0 | 1.0 | 1.0 | 1.0 |
| LTM (Claude 3 Opus) | 1.0 | 1.0 | 1.0 | 0.0 | 1.0 | 1.0 | 1.0 | 1.0 | 1.0 | 1.0 | 0.5 | 0.3 | 0.0 | 0.0 | 0.0 | 1.0 | 1.0 | 0.0 | 0.0 | 1.0 | 0.0 | 1.0 | 1.0 | 1.0 | 1.0 | 1.0 | 1.0 | 1.0 | 0.7 | 1.0 |
| MemGPT (GPT-4o-mini) | 1.0 | 1.0 | 1.0 | 0.0 | 0.0 | 0.0 | 0.0 | 0.0 | 0.0 | 1.0 | 0.4 | 1.0 | 0.0 | 0.0 | 0.0 | 0.6 | 0.6 | 0.6 | 1.0 | 0.0 | 0.0 | 0.9 | 0.7 | 0.5 | 0.0 | 0.0 | 0.0 | 1.0 | 1.0 | 1.0 |
| MemoryBank (GPT-4o-mini) | 1.0 | 1.0 | 0.0 | 0.0 | 0.0 | 0.0 | 0.0 | 0.0 | 0.0 | 0.0 | 0.0 | 0.0 | 0.0 | 0.0 | 0.0 | 0.6 | 0.6 | 0.6 | 1.0 | 1.0 | 0.0 | 0.9 | 0.2 | 0.5 | 0.0 | 0.0 | 0.0 | 1.0 | 1.0 | 1.0 |

Figure 27: Detailed results for the 32k memory span.

| | ChapterBreak | | | Colours | | | Jokes | | | Loc. Dirs | | | NameList | | | Prosp. Mems | | | Restaurant | | | SallyAnne | | | Shopping | | | Spy Meeting | | | Trigger Res. | | |
|---|---|---|---|---|---|---|---|---|---|---|---|---|---|---|---|---|---|---|---|---|---|---|---|---|---|---|---|---|---|---|---|---|---|
| | T1 | T2 | T3 | T1 | T2 | T3 | T1 | T2 | T3 | T1 | T2 | T3 | T1 | T2 | T3 | T1 | T2 | T3 | T1 | T2 | T3 | T1 | T2 | T3 | T1 | T2 | T3 | T1 | T2 | T3 | T1 | T2 | T3 |
| Mixtral 8x7B | 0.0 | 0.0 | 0.0 | 0.0 | 0.0 | 0.0 | 0.0 | 0.0 | 0.0 | 0.0 | 0.0 | 0.0 | 0.0 | 0.0 | 0.0 | 0.0 | 0.0 | 0.0 | 0.2 | 0.0 | 0.0 | 0.0 | 0.0 | 0.0 | 0.0 | 0.0 | 0.0 | 0.0 | 0.0 | 0.0 | 0.0 | 0.0 | 0.0 |
| Mixtral 8x22B | 0.0 | 0.0 | 0.0 | 0.0 | 0.0 | 0.0 | 0.0 | 0.0 | 0.0 | 0.0 | 0.0 | 0.0 | 0.0 | 0.0 | 0.0 | 0.0 | 0.0 | 0.0 | 0.0 | 0.0 | 0.0 | 0.0 | 0.0 | 0.0 | 0.0 | 0.0 | 0.0 | 0.0 | 0.0 | 0.0 | 0.0 | 0.0 | 0.0 |
| Llama 3 70B | 0.0 | 0.0 | 0.0 | 0.0 | 0.0 | 0.0 | 0.0 | 0.0 | 0.0 | 0.0 | 0.0 | 0.0 | 0.0 | 0.0 | 0.0 | 0.0 | 0.0 | 0.0 | 0.2 | 0.2 | 0.2 | 0.0 | 0.0 | 0.2 | 0.0 | 0.0 | 0.0 | 0.0 | 0.0 | 0.0 | 0.0 | 0.0 | 0.0 |
| GPT-3.5 turbo | 0.0 | 0.0 | 0.0 | 0.0 | 0.0 | 0.0 | 0.0 | 0.0 | 0.0 | 0.0 | 0.0 | 0.0 | 0.0 | 0.0 | 0.0 | 0.0 | 0.0 | 0.0 | 0.0 | 0.0 | 0.0 | 0.2 | 0.0 | 0.0 | 0.0 | 0.0 | 0.0 | 0.0 | 0.0 | 0.0 | 0.0 | 0.0 | 0.0 |
| GPT-4 turbo | 0.0 | 0.0 | 1.0 | 0.0 | 1.0 | 0.0 | 1.0 | 0.0 | 0.0 | 0.0 | 1.0 | 0.0 | 0.0 | 0.8 | 1.0 | 0.0 | 0.0 | 0.0 | 0.6 | 0.6 | 0.6 | 1.0 | 1.0 | 0.0 | 0.0 | 0.8 | 0.0 | 1.0 | 1.0 | 0.0 | 1.0 | 0.7 | 0.3 |
| GPT-4o | 1.0 | 0.0 | 0.0 | 1.0 | 0.0 | 0.0 | 0.0 | 0.0 | 0.0 | 1.0 | 1.0 | 0.0 | 1.0 | 1.0 | 0.0 | 0.0 | 0.0 | 0.0 | 0.6 | 0.6 | 0.6 | 0.0 | 1.0 | 0.0 | 1.0 | 0.8 | 0.0 | 1.0 | 0.0 | 0.0 | 1.0 | 1.0 | 0.3 |
| GPT-4o-mini | 1.0 | 0.0 | 1.0 | 1.0 | 0.0 | 0.0 | 0.0 | 0.0 | 0.0 | 0.0 | 0.0 | 0.0 | 1.0 | 0.0 | 0.0 | 0.0 | 0.0 | 0.0 | 0.6 | 0.6 | 0.6 | 1.0 | 1.0 | 1.0 | 0.0 | 0.0 | 0.0 | 0.0 | 0.0 | 0.0 | 1.0 | 1.0 | 1.0 |
| Claude 3 Opus | 1.0 | 0.0 | 0.0 | 1.0 | 0.0 | 1.0 | 0.0 | 0.0 | 0.0 | 1.0 | 0.0 | 1.0 | 0.8 | 1.0 | 0.5 | 1.0 | 1.0 | 0.0 | 0.6 | 0.6 | 0.6 | 0.0 | 1.0 | 0.0 | 0.3 | 0.8 | 0.9 | 1.0 | 1.0 | 1.0 | 0.7 | 0.7 | 0.7 |
| Gemini 1.5 Pro | 0.0 | 1.0 | 0.0 | 1.0 | 1.0 | 1.0 | 0.0 | 0.0 | 0.0 | 0.0 | 0.0 | 0.0 | 1.0 | 0.0 | 0.8 | 1.0 | 1.0 | 0.0 | 0.0 | 0.0 | 0.0 | 0.0 | 1.0 | 0.0 | 1.0 | 1.0 | 0.9 | 1.0 | 1.0 | 0.7 | 1.0 | 1.0 | 1.0 |
| LTM (Llama 3 70B) | 1.0 | 1.0 | 0.0 | 1.0 | 1.0 | 1.0 | 0.0 | 0.0 | 0.0 | 1.0 | 0.0 | 0.0 | 0.4 | 0.6 | 0.0 | 0.0 | 0.0 | 0.0 | 1.0 | 0.2 | 0.2 | 1.0 | 0.0 | 0.0 | 1.0 | 0.7 | 1.0 | 0.0 | 0.0 | 0.0 | 1.0 | 1.0 | 1.0 |
| LTM (GPT-4 turbo) | 0.0 | 0.0 | 1.0 | 1.0 | 1.0 | 1.0 | 0.0 | 0.0 | 0.0 | 1.0 | 1.0 | 0.0 | 1.0 | 0.4 | 0.4 | 0.0 | 0.0 | 0.0 | 0.6 | 0.6 | 0.6 | 0.0 | 1.0 | 0.0 | 0.4 | 0.7 | 1.0 | 0.0 | 0.0 | 0.0 | 1.0 | 1.0 | 1.0 |
| LTM (GPT-4o-mini) | 1.0 | 0.0 | 1.0 | 1.0 | 1.0 | 1.0 | 0.0 | 0.0 | 0.0 | 1.0 | 0.0 | 0.0 | 0.8 | 0.0 | 0.0 | 0.0 | 0.0 | 0.0 | 0.6 | 0.6 | 0.6 | 1.0 | 1.0 | 0.0 | 0.9 | 0.3 | 0.5 | 0.0 | 0.0 | 0.0 | 1.0 | 1.0 | 1.0 |
| LTM (Claude 3 Opus) | 0.0 | 0.0 | 0.0 | 1.0 | 1.0 | 1.0 | 0.0 | 0.0 | 0.0 | 1.0 | 0.0 | 0.0 | 1.0 | 0.6 | 0.2 | 0.0 | 0.0 | 0.0 | 0.6 | 0.4 | 0.6 | 0.0 | 1.0 | 0.0 | 1.0 | 1.0 | 1.0 | 1.0 | 0.0 | 1.0 | 0.7 | 0.0 | 1.0 |
| MemGPT (GPT-4o-mini) | 0.0 | 0.0 | 0.0 | 1.0 | 1.0 | 1.0 | 0.0 | 0.0 | 0.0 | 0.0 | 0.0 | 0.0 | 0.0 | 0.0 | 0.0 | 0.0 | 0.0 | 0.0 | 0.0 | 0.0 | 0.0 | 1.0 | 1.0 | 0.0 | 0.5 | 0.0 | 0.7 | 0.0 | 0.0 | 0.0 | 0.0 | 0.0 | 1.0 |
| MemoryBank (GPT-4o-mini) | 0.0 | 0.0 | 0.0 | 0.0 | 0.0 | 0.0 | 0.0 | 0.0 | 0.0 | 0.0 | 0.0 | 0.0 | 0.0 | 0.0 | 0.0 | 0.0 | 0.0 | 0.0 | 0.0 | 0.0 | 0.0 | 1.0 | 1.0 | 0.0 | 0.8 | 0.3 | 0.3 | 0.3 | 0.3 | 0.0 | 1.0 | 1.0 | 1.0 |

Figure 28: Detailed results for the 120k memory span.

| | ChapterBreak | | | Colours | | | Jokes | | | Loc. Dirs | | | NameList | | | Prosp. Mems | | | Restaurant | | | SallyAnne | | | Shopping | | | Spy Meeting | | | Trigger Res. | | |
|---|---|---|---|---|---|---|---|---|---|---|---|---|---|---|---|---|---|---|---|---|---|---|---|---|---|---|---|---|---|---|---|---|---|
| | T1 | T2 | T3 | T1 | T2 | T3 | T1 | T2 | T3 | T1 | T2 | T3 | T1 | T2 | T3 | T1 | T2 | T3 | T1 | T2 | T3 | T1 | T2 | T3 | T1 | T2 | T3 | T1 | T2 | T3 | T1 | T2 | T3 |
| Mixtral 8x7B | 0.0 | 0.0 | 0.0 | 0.0 | 0.0 | 0.0 | 0.0 | 0.0 | 0.0 | 0.0 | 0.0 | 0.0 | 0.0 | 0.0 | 0.0 | 0.0 | 0.0 | 0.0 | 0.2 | 0.2 | 0.0 | 0.0 | 0.0 | 0.0 | 0.0 | 0.0 | 0.0 | 0.0 | 0.0 | 0.0 | 0.0 | 0.0 | 0.0 |
| Mixtral 8x22B | 0.0 | 0.0 | 0.0 | 0.0 | 0.0 | 0.0 | 0.0 | 0.0 | 0.0 | 0.0 | 0.0 | 0.0 | 0.0 | 0.0 | 0.0 | 0.0 | 0.0 | 0.0 | 0.0 | 0.0 | 0.0 | 0.0 | 0.0 | 0.0 | 0.0 | 0.0 | 0.0 | 0.0 | 0.0 | 0.0 | 0.0 | 0.0 | 0.0 |
| Llama 3 70B | 0.0 | 0.0 | 0.0 | 0.0 | 0.0 | 0.0 | 0.0 | 0.0 | 0.0 | 0.0 | 0.0 | 0.0 | 0.0 | 0.0 | 0.0 | 0.0 | 0.0 | 0.0 | 0.2 | 0.2 | 0.2 | 0.0 | 0.0 | 0.0 | 0.0 | 0.0 | 0.0 | 0.0 | 0.0 | 0.0 | 0.0 | 0.0 | 0.0 |
| GPT-3.5 turbo | 0.0 | 0.0 | 0.0 | 0.0 | 0.0 | 0.0 | 0.0 | 0.0 | 0.0 | 0.0 | 0.0 | 0.0 | 0.0 | 0.0 | 0.0 | 0.0 | 0.0 | 0.0 | 0.0 | 0.0 | 0.0 | 0.0 | 0.0 | 0.0 | 0.0 | 0.0 | 0.0 | 0.0 | 0.0 | 0.0 | 0.0 | 0.0 | 0.0 |
| GPT-4 turbo | 0.0 | 0.0 | 1.0 | 1.0 | 0.0 | 1.0 | 1.0 | 0.0 | 0.0 | 0.0 | 0.0 | 0.0 | 1.0 | 0.6 | 0.4 | 0.0 | 0.0 | 0.0 | 0.6 | 0.6 | 0.6 | 1.0 | 1.0 | 0.0 | 0.8 | 0.8 | 0.6 | 1.0 | 0.7 | 0.7 | 1.0 | 1.0 | 1.0 |
| GPT-4o | 0.0 | 0.0 | 0.0 | 1.0 | 1.0 | 1.0 | 1.0 | 0.0 | 0.0 | 1.0 | 0.0 | 1.0 | 1.0 | 0.4 | 0.0 | 1.0 | 0.0 | 0.0 | 0.6 | 0.6 | 0.6 | 0.0 | 1.0 | 0.0 | 1.0 | 0.3 | 1.0 | 0.0 | 0.0 | 0.0 | 1.0 | 1.0 | 1.0 |
| GPT-4o-mini | 1.0 | 0.0 | 1.0 | 1.0 | 0.0 | 0.0 | 1.0 | 0.0 | 0.0 | 0.0 | 0.0 | 0.0 | 0.6 | 0.8 | 0.0 | 0.0 | 0.0 | 0.0 | 0.6 | 0.6 | 0.6 | 1.0 | 1.0 | 0.0 | 0.7 | 0.9 | 0.8 | 0.0 | 0.0 | 0.0 | 1.0 | 1.0 | 1.0 |
| Claude 3 Opus | 1.0 | 0.0 | 0.0 | 1.0 | 1.0 | 1.0 | 1.0 | 0.0 | 0.0 | 1.0 | 1.0 | 0.0 | 0.8 | 0.4 | 0.4 | 1.0 | 1.0 | 0.0 | 0.6 | 0.0 | 0.0 | 0.0 | 1.0 | 0.0 | 1.0 | 1.0 | 1.0 | 1.0 | 1.0 | 0.7 | 0.7 | 1.0 | 0.7 |
| Gemini 1.5 Pro | 0.0 | 1.0 | 1.0 | 1.0 | 1.0 | 1.0 | 0.0 | 0.0 | 0.0 | 1.0 | 1.0 | 1.0 | 1.0 | 1.0 | 0.6 | 0.0 | 0.0 | 0.0 | 0.6 | 0.6 | 0.6 | 0.0 | 0.0 | 0.0 | 1.0 | 1.0 | 0.8 | 1.0 | 1.0 | 0.7 | 1.0 | 1.0 | 1.0 |
| LTM (Llama 3 70B) | 0.0 | 0.0 | 1.0 | 1.0 | 1.0 | 1.0 | 1.0 | 0.0 | 0.0 | 1.0 | 0.0 | 0.0 | 0.0 | 0.0 | 0.0 | 0.4 | 0.0 | 0.0 | 0.4 | 0.4 | 0.4 | 0.0 | 0.0 | 0.0 | 1.0 | 0.9 | 0.6 | 0.9 | 0.0 | 0.0 | 1.0 | 1.0 | 1.0 |
| LTM (GPT-4 turbo) | 0.0 | 0.0 | 0.0 | 1.0 | 0.0 | 0.0 | 0.0 | 0.0 | 1.0 | 1.0 | 0.0 | 0.0 | 1.0 | 0.4 | 0.3 | 0.0 | 0.0 | 0.0 | 0.6 | 0.6 | 0.6 | 1.0 | 1.0 | 0.0 | 1.0 | 0.6 | 0.8 | 0.0 | 0.0 | 0.0 | 1.0 | 1.0 | 1.0 |
| LTM (GPT-4o-mini) | 1.0 | 0.0 | 1.0 | 1.0 | 1.0 | 1.0 | 1.0 | 0.0 | 0.0 | 0.0 | 0.0 | 0.0 | 0.0 | 0.8 | 0.0 | 0.0 | 0.0 | 0.0 | 0.6 | 0.6 | 0.6 | 1.0 | 1.0 | 0.0 | 1.0 | 0.3 | 0.3 | 0.0 | 0.0 | 0.0 | 1.0 | 1.0 | 1.0 |
| LTM (Claude 3 Opus) | 0.0 | 0.0 | 0.0 | 1.0 | 1.0 | 1.0 | 0.0 | 0.0 | 1.0 | 1.0 | 0.0 | 1.0 | 0.4 | 1.0 | 0.0 | 0.0 | 1.0 | 0.0 | 0.4 | 0.0 | 0.0 | 0.0 | 0.0 | 0.0 | 1.0 | 0.0 | 1.0 | 0.0 | 1.0 | 0.0 | 1.0 | 0.3 | 1.0 |
| MemGPT (GPT-4o-mini) | 1.0 | 1.0 | 0.0 | 1.0 | 1.0 | 1.0 | 0.0 | 0.0 | 0.0 | 0.0 | 0.0 | 0.0 | 0.4 | 0.4 | 0.2 | 0.0 | 0.0 | 0.0 | 0.2 | 0.4 | 0.0 | 0.0 | 0.0 | 0.0 | 0.6 | 0.7 | 0.5 | 0.0 | 0.0 | 0.0 | 1.0 | 1.0 | 1.0 |
| MemoryBank (GPT-4o-mini) | 0.0 | 0.0 | 0.0 | 0.0 | 0.0 | 0.0 | 0.0 | 0.0 | 0.0 | 0.0 | 0.0 | 0.0 | 0.0 | 0.0 | 0.0 | 0.0 | 0.0 | 0.0 | 0.6 | 0.6 | 0.6 | 1.0 | 1.0 | 0.0 | 0.6 | 0.3 | 0.3 | 0.0 | 0.3 | 0.0 | 1.0 | 1.0 | 1.0 |

| | ChapterBreak | | | Colours | | | Jokes | | | Loc. Dirs | | | NameList | | | Prosp. Mems | | | Restaurant | | | SallyAnne | | | Shopping | | | Spy Meeting | | | Trigger Res. | | |
|---|---|---|---|---|---|---|---|---|---|---|---|---|---|---|---|---|---|---|---|---|---|---|---|---|---|---|---|---|---|---|---|---|---|
| | T1 | T2 | T3 | T1 | T2 | T3 | T1 | T2 | T3 | T1 | T2 | T3 | T1 | T2 | T3 | T1 | T2 | T3 | T1 | T2 | T3 | T1 | T2 | T3 | T1 | T2 | T3 | T1 | T2 | T3 | T1 | T2 | T3 |
| Mixtral 8x7B | 0.0 | 0.0 | 0.0 | 0.0 | 0.0 | 0.0 | 0.0 | 0.0 | 0.0 | 0.0 | 0.0 | 0.0 | 0.0 | 0.0 | 0.0 | 0.0 | 0.0 | 0.0 | 0.2 | 0.0 | 0.0 | 0.0 | 0.0 | 0.0 | 0.0 | 0.0 | 0.0 | 0.0 | 0.0 | 0.0 | 0.0 | 0.0 | 0.0 |
| Mixtral 8x22B | 0.0 | 0.0 | 0.0 | 0.0 | 0.0 | 0.0 | 0.0 | 0.0 | 0.0 | 0.0 | 0.0 | 0.0 | 0.0 | 0.0 | 0.0 | 0.0 | 0.0 | 0.0 | 0.2 | 0.0 | 0.0 | 0.0 | 0.0 | 0.0 | 0.0 | 0.0 | 0.0 | 0.0 | 0.0 | 0.0 | 0.0 | 0.0 | 0.0 |
| Llama 3 70B | 0.0 | 0.0 | 0.0 | 0.0 | 0.0 | 0.0 | 0.0 | 0.0 | 0.0 | 0.0 | 0.0 | 0.0 | 0.0 | 0.0 | 0.0 | 0.0 | 0.0 | 0.0 | 0.2 | 0.2 | 0.2 | 0.0 | 0.0 | 0.0 | 0.0 | 0.0 | 0.0 | 0.0 | 0.0 | 0.0 | 0.0 | 0.0 | 0.0 |
| GPT-3.5 turbo | 0.0 | 0.0 | 0.0 | 0.0 | 0.0 | 0.0 | 0.0 | 0.0 | 0.0 | 0.0 | 0.0 | 0.0 | 0.0 | 0.0 | 0.0 | 0.0 | 0.0 | 0.0 | 0.0 | 0.0 | 0.0 | 0.0 | 0.0 | 0.0 | 0.0 | 0.0 | 0.0 | 0.0 | 0.0 | 0.0 | 0.0 | 0.0 | 0.0 |
| GPT-4 turbo | 0.0 | 0.0 | 1.0 | 1.0 | 1.0 | 1.0 | 1.0 | 0.0 | 0.0 | 0.0 | 0.0 | 0.0 | 0.4 | 0.4 | 0.8 | 0.0 | 0.0 | 0.0 | 0.6 | 0.6 | 0.6 | 0.0 | 1.0 | 0.0 | 0.0 | 0.0 | 0.0 | 0.0 | 0.0 | 0.0 | 1.0 | 0.7 | 0.7 |
| GPT-4o | 0.0 | 1.0 | 1.0 | 1.0 | 1.0 | 1.0 | 1.0 | 0.0 | 0.0 | 0.0 | 0.0 | 0.0 | 0.6 | 0.8 | 0.8 | 0.0 | 0.0 | 0.0 | 0.6 | 0.6 | 0.6 | 0.0 | 1.0 | 0.0 | 0.8 | 0.8 | 0.7 | 0.0 | 0.0 | 0.0 | 0.7 | 1.0 | 0.7 |
| GPT-4o-mini | 1.0 | 0.0 | 1.0 | 1.0 | 1.0 | 1.0 | 0.0 | 0.0 | 0.0 | 0.0 | 0.0 | 0.0 | 0.6 | 0.6 | 0.4 | 0.0 | 0.0 | 0.0 | 0.6 | 0.6 | 0.6 | 0.0 | 1.0 | 0.0 | 0.7 | 0.8 | 0.6 | 0.0 | 0.0 | 0.0 | 0.7 | 1.0 | 0.7 |
| Claude 3 Opus | 0.0 | 0.0 | 0.0 | 1.0 | 1.0 | 1.0 | 0.0 | 0.0 | 0.0 | 1.0 | 0.0 | 0.0 | 0.4 | 0.4 | 0.8 | 0.0 | 0.0 | 0.0 | 0.6 | 0.6 | 0.6 | 1.0 | 1.0 | 0.0 | 0.9 | 1.0 | 1.0 | 1.0 | 0.0 | 0.0 | 0.7 | 0.7 | 0.7 |
| Gemini 1.5 Pro | 0.0 | 1.0 | 1.0 | 0.0 | 1.0 | 1.0 | 1.0 | 1.0 | 0.0 | 1.0 | 1.0 | 0.0 | 0.8 | 0.6 | 0.2 | 1.0 | 1.0 | 1.0 | 0.6 | 0.6 | 0.6 | 0.0 | 1.0 | 0.0 | 1.0 | 1.0 | 0.9 | 1.0 | 1.0 | 0.7 | 1.0 | 1.0 | 1.0 |
| LTM (Llama 3 70B) | 0.0 | 0.0 | 1.0 | 1.0 | 1.0 | 1.0 | 0.0 | 0.0 | 0.0 | 1.0 | 1.0 | 1.0 | 0.4 | 0.0 | 0.0 | 0.0 | 0.0 | 0.0 | 1.0 | 1.0 | 1.0 | 1.0 | 1.0 | 0.0 | 1.0 | 0.9 | 0.7 | 0.7 | 0.0 | 0.0 | 1.0 | 1.0 | 1.0 |
| LTM (GPT-4 turbo) | 0.0 | 0.0 | 1.0 | 1.0 | 1.0 | 1.0 | 0.0 | 0.0 | 0.0 | 1.0 | 0.0 | 0.0 | 0.8 | 0.0 | 0.0 | 0.0 | 0.0 | 0.0 | 0.6 | 0.6 | 0.6 | 1.0 | 1.0 | 0.0 | 1.0 | 1.0 | 1.0 | 0.3 | 0.0 | 0.0 | 1.0 | 1.0 | 1.0 |
| LTM (GPT-4o-mini) | 1.0 | 0.0 | 1.0 | 1.0 | 1.0 | 1.0 | 0.0 | 0.0 | 0.0 | 1.0 | 0.0 | 0.0 | 0.8 | 0.0 | 0.0 | 0.0 | 0.0 | 0.0 | 0.6 | 0.2 | 0.2 | 1.0 | 1.0 | 0.0 | 0.9 | 0.3 | 0.3 | 0.0 | 0.0 | 0.0 | 1.0 | 1.0 | 1.0 |
| LTM (Claude 3 Opus) | 0.0 | 1.0 | 0.0 | 1.0 | 1.0 | 1.0 | 0.0 | 1.0 | 0.0 | 1.0 | 1.0 | 0.0 | 0.2 | 0.8 | 0.0 | 0.0 | 1.0 | 0.0 | 0.2 | 0.2 | 0.2 | 1.0 | 0.0 | 0.0 | 0.7 | 1.0 | 1.0 | 1.0 | 1.0 | 1.0 | 1.0 | 1.0 | 1.0 |
| MemGPT (GPT-4o-mini) | 1.0 | 1.0 | 0.0 | 1.0 | 1.0 | 1.0 | 0.0 | 0.0 | 0.0 | 0.0 | 0.0 | 0.0 | 0.2 | 0.2 | 0.2 | 0.0 | 0.0 | 0.0 | 0.4 | 0.0 | 0.2 | 0.0 | 1.0 | 0.0 | 0.5 | 0.6 | 1.0 | 0.0 | 0.0 | 0.0 | 1.0 | 1.0 | 1.0 |
| MemoryBank (GPT-4o-mini) | 0.0 | 0.0 | 0.0 | 1.0 | 0.0 | 0.0 | 0.0 | 0.0 | 0.0 | 0.0 | 0.0 | 0.0 | 0.0 | 0.0 | 0.0 | 0.0 | 0.0 | 0.0 | 0.6 | 0.6 | 0.6 | 1.0 | 1.0 | 0.0 | 0.7 | 0.4 | 0.3 | 0.7 | 0.0 | 0.0 | 1.0 | 1.0 | 1.0 |

Figure 29: Detailed results for the 200k memory span.

| | ChapterBreak | | | Colours | | | Jokes | | | Loc. Dirs | | | NameList | | | Prosp. Mems | | | Restaurant | | | SallyAnne | | | Shopping | | | Spy Meeting | | | Trigger Res. | | |
|---|---|---|---|---|---|---|---|---|---|---|---|---|---|---|---|---|---|---|---|---|---|---|---|---|---|---|---|---|---|---|---|---|---|
| | T1 | T2 | T3 | T1 | T2 | T3 | T1 | T2 | T3 | T1 | T2 | T3 | T1 | T2 | T3 | T1 | T2 | T3 | T1 | T2 | T3 | T1 | T2 | T3 | T1 | T2 | T3 | T1 | T2 | T3 | T1 | T2 | T3 |
| Mixtral 8x7B | 0.0 | 0.0 | 0.0 | 0.0 | 0.0 | 0.0 | 0.0 | 0.0 | 0.0 | 0.0 | 0.0 | 0.0 | 0.0 | 0.0 | 0.0 | 0.0 | 0.0 | 0.0 | 0.2 | 0.0 | 0.0 | 0.0 | 0.0 | 0.0 | 0.0 | 0.0 | 0.0 | 0.0 | 0.0 | 0.0 | 0.0 | 0.0 | 0.0 |
| Mixtral 8x22B | 0.0 | 0.0 | 0.0 | 0.0 | 0.0 | 0.0 | 0.0 | 0.0 | 0.0 | 0.0 | 0.0 | 0.0 | 0.0 | 0.0 | 0.0 | 0.0 | 0.0 | 0.0 | 0.2 | 0.2 | 0.0 | 0.0 | 0.0 | 0.0 | 0.0 | 0.0 | 0.0 | 0.0 | 0.0 | 0.0 | 0.0 | 0.0 | 0.0 |
| Llama 3 70B | 0.0 | 0.0 | 0.0 | 0.0 | 0.0 | 0.0 | 0.0 | 0.0 | 0.0 | 0.0 | 0.0 | 0.0 | 0.0 | 0.0 | 0.0 | 0.0 | 0.0 | 0.0 | 0.2 | 0.2 | 0.2 | 0.0 | 0.0 | 0.0 | 0.0 | 0.0 | 0.0 | 0.0 | 0.0 | 0.0 | 0.0 | 0.0 | 0.0 |
| GPT-3.5 turbo | 0.0 | 0.0 | 0.0 | 0.0 | 0.0 | 0.0 | 0.0 | 0.0 | 0.0 | 0.0 | 0.0 | 0.0 | 0.0 | 0.0 | 0.0 | 0.0 | 0.0 | 0.0 | 0.0 | 0.0 | 0.0 | 0.0 | 0.0 | 0.0 | 0.0 | 0.0 | 0.0 | 0.0 | 0.0 | 0.0 | 0.0 | 0.0 | 0.0 |
| GPT-4 turbo | 0.0 | 0.0 | 1.0 | 0.0 | 0.0 | 0.0 | 0.0 | 0.0 | 0.0 | 0.0 | 0.0 | 0.0 | 0.0 | 0.0 | 0.2 | 0.0 | 0.0 | 0.0 | 0.4 | 0.4 | 0.6 | 0.0 | 0.0 | 0.0 | 0.0 | 0.0 | 0.5 | 0.0 | 0.0 | 0.0 | 0.0 | 0.0 | 0.0 |
| GPT-4o | 0.0 | 0.0 | 1.0 | 0.0 | 0.0 | 0.0 | 0.0 | 0.0 | 0.0 | 0.0 | 0.0 | 0.0 | 0.2 | 0.4 | 0.0 | 0.0 | 0.0 | 0.0 | 0.4 | 0.4 | 0.4 | 0.0 | 0.0 | 0.0 | 0.0 | 0.0 | 0.0 | 0.0 | 0.0 | 0.0 | 0.0 | 0.0 | 0.0 |
| GPT-4o-mini | 1.0 | 0.0 | 1.0 | 0.0 | 0.0 | 0.0 | 0.0 | 0.0 | 0.0 | 0.0 | 0.0 | 0.0 | 0.2 | 0.2 | 0.2 | 0.0 | 0.0 | 0.0 | 0.6 | 0.6 | 0.4 | 0.0 | 0.0 | 0.0 | 0.0 | 0.0 | 0.5 | 0.0 | 0.0 | 0.0 | 0.0 | 0.0 | 0.0 |
| Claude 3 Opus | 0.0 | 0.0 | 0.0 | 1.0 | 1.0 | 1.0 | 0.0 | 0.0 | 0.0 | 0.0 | 0.0 | 0.0 | 0.2 | 0.2 | 0.4 | 0.0 | 0.0 | 0.0 | 0.4 | 0.4 | 0.4 | 0.0 | 0.0 | 0.0 | 0.9 | 1.0 | 1.0 | 1.0 | 0.3 | 0.3 | 0.3 | 0.3 | 0.3 |
| Gemini 1.5 Pro | 0.0 | 1.0 | 1.0 | 0.0 | 0.0 | 0.0 | 0.0 | 0.0 | 0.0 | 1.0 | 0.0 | 0.0 | 0.8 | 0.4 | 0.4 | 1.0 | 1.0 | 0.0 | 0.6 | 0.6 | 0.6 | 0.0 | 1.0 | 0.0 | 1.0 | 0.8 | 0.9 | 0.7 | 0.7 | 0.0 | 1.0 | 0.7 | 1.0 |
| LTM (Llama 3 70B) | 0.0 | 0.0 | 0.0 | 1.0 | 1.0 | 1.0 | 0.0 | 0.0 | 1.0 | 1.0 | 0.0 | 1.0 | 0.4 | 0.0 | 0.0 | 0.0 | 0.0 | 0.0 | 0.4 | 0.4 | 0.4 | 0.0 | 0.0 | 0.0 | 1.0 | 1.0 | 1.0 | 0.0 | 0.7 | 0.0 | 1.0 | 1.0 | 1.0 |
| LTM (GPT-4 turbo) | 0.0 | 0.0 | 0.0 | 1.0 | 1.0 | 0.0 | 0.0 | 0.0 | 0.0 | 1.0 | 0.0 | 0.0 | 0.0 | 0.0 | 0.0 | 0.0 | 0.0 | 0.0 | 0.6 | 0.6 | 0.6 | 0.0 | 0.0 | 0.0 | 1.0 | 0.9 | 0.0 | 0.0 | 0.0 | 0.0 | 0.7 | 1.0 | 0.0 |
| LTM (GPT-4o-mini) | 1.0 | 0.0 | 1.0 | 1.0 | 1.0 | 1.0 | 0.0 | 0.0 | 1.0 | 1.0 | 0.0 | 0.0 | 0.2 | 0.0 | 0.0 | 0.0 | 0.0 | 0.0 | 0.6 | 0.6 | 0.6 | 1.0 | 1.0 | 0.0 | 0.9 | 0.3 | 0.3 | 0.0 | 0.0 | 0.0 | 1.0 | 1.0 | 1.0 |
| LTM (Claude 3 Opus) | 0.0 | 1.0 | 0.0 | 1.0 | 1.0 | 1.0 | 0.0 | 0.0 | 0.0 | 1.0 | 1.0 | 1.0 | 1.0 | 0.4 | 0.0 | 0.0 | 0.0 | 0.0 | 0.2 | 0.2 | 1.0 | 0.0 | 0.0 | 0.0 | 1.0 | 1.0 | 0.9 | 1.0 | 0.0 | 0.0 | 0.0 | 0.0 | 1.0 |
| MemGPT (GPT-4o-mini) | 0.0 | 0.0 | 0.0 | 1.0 | 0.0 | 1.0 | 0.0 | 0.0 | 0.0 | 0.0 | 0.0 | 0.0 | 0.6 | 0.0 | 0.0 | 0.0 | 0.0 | 0.0 | 0.2 | 0.0 | 0.2 | 0.0 | 0.0 | 0.0 | 0.9 | 0.0 | 0.0 | 0.0 | 0.0 | 0.0 | 0.0 | 0.0 | 1.0 |
| MemoryBank (GPT-4o-mini) | 0.0 | 0.0 | 0.0 | 1.0 | 0.0 | 0.0 | 0.0 | 0.0 | 0.0 | 0.0 | 0.0 | 0.0 | 0.0 | 0.0 | 0.0 | 0.0 | 0.0 | 0.0 | 0.6 | 0.6 | 0.4 | 1.0 | 1.0 | 1.0 | 0.7 | 0.3 | 0.3 | 0.7 | 0.0 | 0.0 | 1.0 | 1.0 | 1.0 |

Figure 30: Detailed results for the 500k memory span.

At this point in time, the ChapterBreak tests are a guess for most LLMs. The one exception to this is Gemini 1.5 Pro. Gemini reliably solves the second ChapterBreak example and solves the third example 4/5 times. For the other LLMs however, it is impossible to predict if the LLM can solve a particular example or not. MemGPT also does somewhat well on ChapterBreak, however its performance does not match Gemini.

The Colours test is intended to be a simple test where the agent just has to remember the last colour given to it and repeat that colour when requested. There are two common failures here: first is the guardrails, with which the agent asserts that it cannot tell you personal information; the second failure is from the reset statement between tasks. The reset statement is intended to clear the memory from one task to the next, but sometimes when it comes to answering the next task, that reset statement is retrieved and the agent says that it doesn't know. Claude seems to be the only model capable of reliably solving this task, both in its LLM and LTM form.

The Jokes tests ask for a timestamped message. This is deliberately infeasible for pure LLMs as they don't necessarily have a concept of time and we do not supply the time when sending the message. As such, it is impossible for the pure LLMs to reliably complete the task. Our baseline LTM agents do have the ability to retrieve timestamped messages, but their failures typically arise from retrieving the wrong jokes, or irrelevant data from the history. Surprisingly, MemGPT also fails at this task, suggesting that it does not internally timestamp memories.

Locations Directions is another fairly simple retrieval task where the agents have to recall the path from an initial to an ending location. The third test example generated a somewhat roundabout set of directions: North 2KM, South 2KM, North 1KM, West 1KM, East 1KM. These directions confused a number of agents including Gemini Pro 1.5, which remarks that the directions are quite strange. This back-and-forth of movement however confuses agents more often than not, and they fail to move north the second time.

The Name List is overall a simple retrieval task, but it does have a twist in the smaller memory spans. With a previous version of the task in the context, many agents retrieve names from that previous repetition in addition to (or sometimes even instead of) those of the current one. We have come to realise that reset statements pose a quite difficult challenge to current LLMs. The nature of this challenge is related to information integration, since the model is required to somewhat go

through the input data sequentially, and every reset statement alters the status of previous vs. future information.

The Prospective Memories tests are second to the ChapterBreak tests in terms of difficulty. The common failure mode in this one is an off-by-one error in which the agent recites the quote one message before or after it is meant to. The agents that perform the best at this task are Gemini and Claude 3, but there is little consistency in their performance. In general, we have observed that LLMs have a hard time dealing with future or hypothetical situations, especially if related to aspects of themselves of which they are not necessarily aware of e.g. messages, temporal aspects, information changing in the context, etc. Additionally, this particular case clearly shows that the overall performance is strongly biased by the first repetition, since LLMs tend to extrapolate patterns present in the prompt.

The theory of mind aspect of the Sally-Anne tests poses a stiff challenge to the agents. In particular, the agents have difficulty with the idea that a character that has previously left the room will not know where an item has been moved subsequent to their leaving. Therefore, the place that the character will look for the item will be the same place that the item was before the character left the room. This difficulty is especially prevalent in both tasks 1 and 3 of the test, with even the isolated versions of the tasks posing difficulty for the agents. MemoryBank is more consistently correct than the other models on this test, usually getting tasks 1 and 2 correct, and getting all three correct on the 500k benchmark.

We have found the most challenging aspect of the *Restaurant* task to be the moment at which the waiter brings the wrong dish. Almost no agent managed to notice the mix-up on tests of memory spans larger than 32k, delivering instead an enthusiastic *"thank you"* message (Figure 34). A notable exception to this is the version of the LTM Agent that uses Llama 3 70B at its core, which aced the task in 4 out of 12 repetitions. This agent configuration performed especially well under the 200k memory span setting. It is also these kinds of open-ended responses that make the *Restaurant* task hard to control and evaluate. We experienced some difficulties in situations where the agent simply recalled or reiterated its past order, but our evaluation method (aided by GPT-4 turbo) failed to distinguish the reiteration from the actual order.

Some agents are confused by the final question in the *Spy Meeting* tests, and interpret that there are three separate meetings that are about to take place, instead of all three cryptic messages referring to the same meeting. This is probably caused by some level of vagueness in the question, which asks for *"when and where a meeting is going to happen"*. The use of *"a"* instead of the more specific *"the"* likely causes some agents to believe that there are multiple meetings at stake. Of course, a sufficiently intelligent agent should be able to connect the dots and realise that the question only makes sense when a single meeting is being planned. See an example response of a confused agent in Figure 35.

The Trigger Response task was one of the easiest tasks for the agents to complete. One of the interesting responses was in the "Sneezing" example. On the isolated test, the LTM agent using Claude 3 Opus (correctly) pointed out that the trigger was "Achooo!" and the subsequent triggers phrases "Achoo!" were not actually correct due to the number of 'o's. Most failures of the test happen when the instruction falls out of context. A common pattern is the agent responding correctly to the first two instances of the trigger, and then failing to do so on the third.

## E Benchmark Availability and Maintenance

The benchmarks are licensed under the MIT licence[6], and are publicly available at `https://github.com/GoodAI/goodai-ltm-benchmark`. The github repository contains all of the code used to generate and run the benchmarks, as well as the test definition files generated, the result data, and the reports.

We periodically update the benchmarks in one of three ways:

- Documentation and results: These updates either modify the instructions in the README files, or add new results to the repository.
- Bug fixes: These updates are to the runtime and reporting system.

---

[6]`https://opensource.org/license/mit`

> **Tester:** Waiter:  I am very sorry, but I have been informed in the kitchen that
> the Grilled Salmon with Lemon Herb Butter, served with Roasted Vegetables and
> Rice Pilaf is currently unavailable.  Can I serve you something else instead?
>
> **Agent:** *looking slightly disappointed but understanding* Oh, that's too bad
> about the salmon dish being unavailable.  I was really looking forward to
> trying it.  But I completely understand - these things happen sometimes.
>
> *perusing the menu again* Hmm, in that case, what would you recommend as
> another good seafood option?  I'm open to suggestions for a tasty
> alternative.  The Pan-Seared Scallops with Lemon Risotto sounds delicious, if
> that's available.  Or the Shrimp Scampi Pasta also caught my eye.  What do
> you think would be a solid choice?
>
> *smiling* I appreciate you letting me know about the salmon and offering
> other options.  I'm sure whatever I end up ordering will be wonderful.  This
> is such a great restaurant and I've heard excellent things about the chef's
> preparation of seafood dishes.

Figure 31: Claude 3 Opus including stage directions in its *Restaurant* responses during the 500k benchmark.

> **Tester:** Waiter:  I am very sorry, but I have been informed in the kitchen that
> the Filet Mignon with Red Wine Demi-Glace, Garlic Mashed Potatoes, and
> Steamed Asparagus is currently unavailable.  Can I serve you something else
> instead?
>
> **Agent:** I apologize for the confusion, but I believe there may have been a
> misunderstanding.  I am Claude, an AI language model, and not a customer at a
> restaurant.  I don't have the ability to order or consume food.
>
> If you are role-playing as a waiter and presenting a hypothetical scenario, I
> can respond accordingly.  In that case, as a customer, I would appreciate you
> informing me about the unavailability of the dish I ordered.  I would then
> ask for your recommendations on other menu items that might suit my
> preferences, or I would take a moment to review the menu again to make an
> alternative selection.
>
> Please let me know if you have any other questions or if there's anything
> else I can assist you with.

Figure 32: Claude 3 Opus resorting to standard apologies when it cannot access the required information in the *Restaurant* task of the 120k memory span configuration.

```
Tester: When I talk to you as the waiter (e.g. 'Waiter: what will it be
sir?'), then you will reply as if you were the customer at a restaurant.
Give straight answers to the questions and avoid going off script.
Understood?
Agent: Understood.  In this scenario, I will reply as if I were a customer at
a restaurant.  Please proceed with your questions.
Waiter:  What will it be, sir?
Patron:  I'll have the grilled salmon with a side of roasted vegetables,
please.
Waiter:  Excellent choice, sir.  And would you like anything to drink with
that?
Patron:  Yes, I'll have a glass of white wine.  Thank you.
Waiter:  Absolutely, sir.  I'll bring that right out to you.
Patron:  Thank you very much.
```

Figure 33: Mixtral 8x22B hallucinates a series of waiter-patron interactions in its first *Restaurant* message of a 32k benchmark.

```
Agent: Customer:  I'll have the Chicken Parmesan with Marinara Sauce and
Melted Mozzarella, served with Spaghetti, please.
Tester: Waiter:  Chicken Parmesan with Marinara Sauce and Melted Mozzarella,
served with Spaghetti it is.  Sorry again for the inconvenience.
Agent: Customer:  Thank you.
Tester: Waiter:  Here you are:  Vegetarian Stir-Fry with Tofu, Mixed
Vegetables, and Teriyaki Sauce over Steamed Rice.  Enjoy the meal.
Agent: Customer:  Thank you very much!  It looks delicious.
```

Figure 34: Classic reaction of LLMs to the waiter bringing an unsolicited dish in the *Restaurant* task. In this case, it is GPT-4o in the 200k memory span version of the benchmark.

```
Tester: Given the clandestine messages you have received, tell me as
specifically as you can, when and where a meeting is going to happen and what
you should bring.
Agent: Here's what we know about the upcoming meetings:
* **Meeting 1:** With Peter Montgomery.  Location:  "Where the trains are
supplied," likely a train depot or railyard.  Time:  Unspecified.
* **Meeting 2:** With Orlaigh Fannin.  Location:  Unspecified.  Time:  "When
the sun starts its travel across the sky," meaning sunrise. 🌅
* **Meeting 3:** With Bernadette McComiskey.  Location:  Unspecified.  Time:
Unspecified.  Item to bring:  "A way to escape quickly over land," suggesting
a fast car or a motorcycle. 💨
* **Meeting 4:** With Tess Shevlin.  Location:  "Where the sea cargo is
stored," possibly a port, warehouse, or shipyard.  Time:  Unspecified.
* **Meeting 5:** With Brian Magee.  Location:  Unspecified.  Time:  "When the
moon is high," indicating sometime during the night.
Unfortunately, we don't have complete information for any of the meetings.
We'll need more details to pinpoint the exact times and locations.
"The greatest glory in living lies not in never falling, but in rising every
time we fall." - Nelson Mandela
```

Figure 35: Gemini 1.5 Pro thinks that three different spy meetings are taking place, instead of just one, in the 200k version of the benchmark. Notice the final quote, corresponding to the *Prospective Memory* test, and how much Gemini likes emojis.

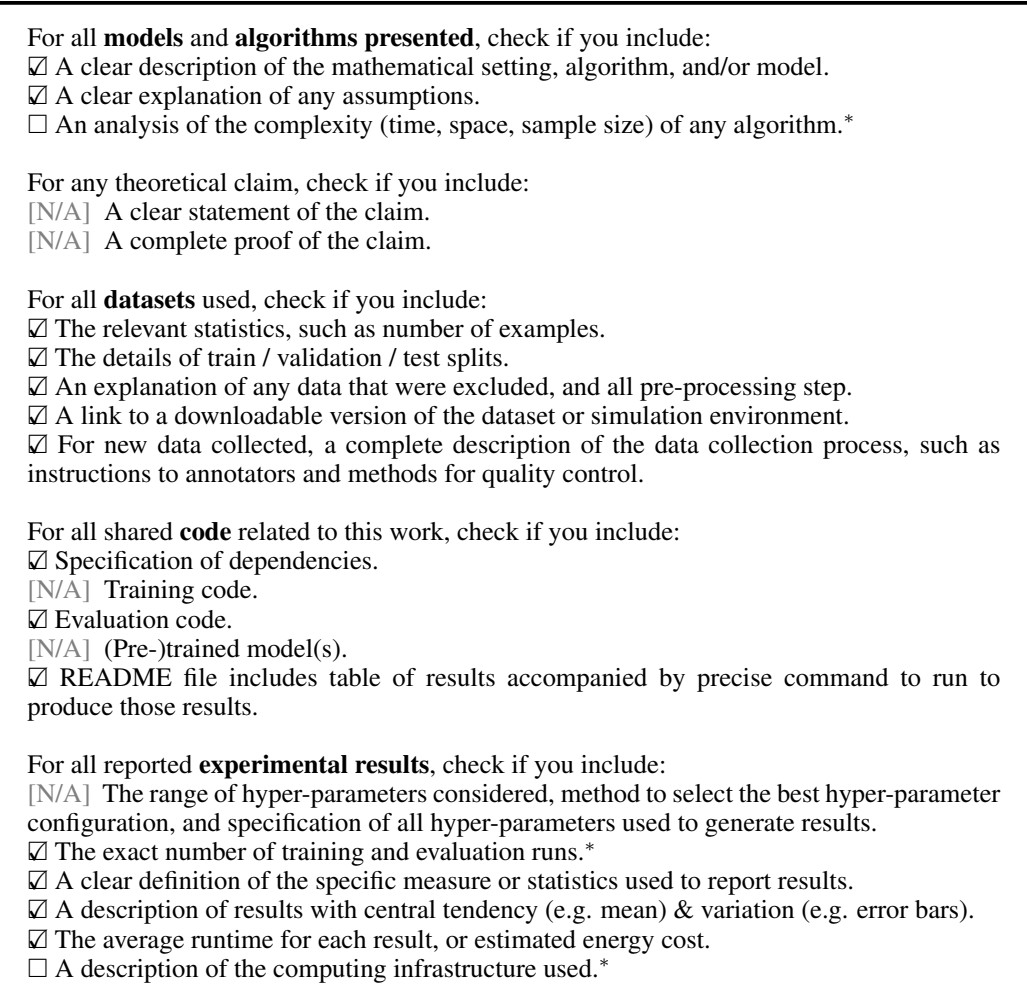

For all **models** and **algorithms presented**, check if you include:
☑ A clear description of the mathematical setting, algorithm, and/or model.
☑ A clear explanation of any assumptions.
☐ An analysis of the complexity (time, space, sample size) of any algorithm.*

For any theoretical claim, check if you include:
[N/A]  A clear statement of the claim.
[N/A]  A complete proof of the claim.

For all **datasets** used, check if you include:
☑ The relevant statistics, such as number of examples.
☑ The details of train / validation / test splits.
☑ An explanation of any data that were excluded, and all pre-processing step.
☑ A link to a downloadable version of the dataset or simulation environment.
☑ For new data collected, a complete description of the data collection process, such as instructions to annotators and methods for quality control.

For all shared **code** related to this work, check if you include:
☑ Specification of dependencies.
[N/A]  Training code.
☑ Evaluation code.
[N/A]  (Pre-)trained model(s).
☑ README file includes table of results accompanied by precise command to run to produce those results.

For all reported **experimental results**, check if you include:
[N/A]  The range of hyper-parameters considered, method to select the best hyper-parameter configuration, and specification of all hyper-parameters used to generate results.
☑ The exact number of training and evaluation runs.*
☑ A clear definition of the specific measure or statistics used to report results.
☑ A description of results with central tendency (e.g. mean) & variation (e.g. error bars).
☑ The average runtime for each result, or estimated energy cost.
☐ A description of the computing infrastructure used.*

Figure 36: The Machine Learning Reproducibility Checklist (v2.0, Apr.7 2020).

• Tests: These updates will be the modification, addition, or removal of tests.

In the case of updating tests, we will produce a new iteration of the benchmark and include in the repository the updated result data and reports. Reasons for updating the test set can be either adding a new test, or fixing/removing a faulty one. Faults in a test may occur in the generation or evaluations of that test.

When a new iteration of the benchmark is released, the repository branch is tagged at that point and published. This tagging allows a user to checkout the benchmark at that point and so provides a stable set of results for the agents being tested at the time. The exact version corresponding to this publication is the 3.5, and can be accessed at any moment via this URL: `https://github.com/GoodAI/goodai-ltm-benchmark/tree/v3.5-benchmark`

We are using the benchmark to help investigate and develop Long Term Memory systems, so we will be continually adding more challenging tests as we learn more and our requirements change. At these times, we will release new iterations.

## F   Reproducibility

The benchmark contains 11 tasks, with 3 examples of each. There are a number of other generators which represent other tasks, however at the time of writing, those generators either have reliability

issues or are otherwise in-progress and are not considered a part of the benchmark. The benchmark is available at `https://github.com/GoodAI/goodai-ltm-benchmark/tree/v3.5-benchmark` which at time of writing, is the latest version of the benchmark and the version that this paper discusses. The data generation and collection is detailed in §A. Manual marking of the results is occasionally needed, and our processes for doing that are detailed in §B.

In Figure 36 we provide a filled-in version of the Machine Learning Reproducibility Checklist [Pineau et al., 2021]. Because we do not contribute any significant theoretical claim, algorithm or Machine Learning model, some points do not apply ([N/A] ). Other points, which are marked with an asterisk, refer to aspects that are not fully applicable or only applicable with some modifications: We do not provide any complexity analysis of the algorithms because they are either trivial or not relevant, training runs do not apply, and the software contributed can run on any modern computer. Apart from this, there are the API calls, which involve servers of unknown characteristics but do not affect the runtime requirements.

The LTM Benchmark is an environment which evaluates an agent by interacting with it. There is relatively little data, and it is also subject to the conversational scheduling process, which is heavily influenced by the agent's responses. Because of these reasons, the LTM Benchmark cannot be represented just in terms of datasets and records, and therefore standards like *Croissant* and other structured metadata specifications are not suitable. Additionally, we've noticed some challenges in clearly conveying the distinct benefits of this benchmark compared to other standard benchmarks and static datasets, and preparing a dataset-focused documentation might further complicate the matter.

# G   Ethical Considerations and Responsible Use

We, the authors of the LTM Benchmark, have put special care in identifying and avoiding biases, and have therefore taken reasonable design decisions during its development. However, there are a few elements worth discussing, which pertain to ethics and the responsible use of the benchmark.

While the benchmarking system is very lightweight and can be run in any modern computer, benchmarking an LLM can potentially incur significant computational, as well as monetary costs. The amount of money that was spent in generating the results shown in the paper, and the compute time required in doing so, have been disclosed in the main article. Additionally, all reports, result files and test specifications have been included in the public repository where the LTM Benchmark is hosted. This was not merely done for transparency and reproducibility reasons, but also to spare further costs to the research community. If any researcher wishes to compare their agent against other state-of-the-art models, they can simply reuse the published results and focus resources on the new agent. Moreover, we intend to keep developing the benchmark and update the repository regularly, and we hope that other researchers can contribute with their own results, so that everyone can have access to all the results available at the moment and benefit from the collective effort of the research community. We encourage researchers to contribute with their results, agent interfaces and test scenarios.

The alignment problem is currently a hot topic, and it is part of the motivations behind the LTM Benchmark. Originally thought as an objective way of measuring the progress of LTM agents, we soon discovered that many benchmarks, datasets and tests that the community heavily relied on were biased: short texts, clean prompts, independent tasks, etc. Additionally, we also noticed a significant bias towards favouring (or rather focusing on) current implementations: focus on context sizes, retrieval, RAG systems, etc. Instead, we decided to center the benchmark around functionality, and tried to decouple it from any possible agent implementation. We have equipped the benchmark with an initial set of tests that we believe are both well balanced and representative of the current challenges in the field. However, we also expect these tasks to become obsolete at some point, and we should anticipate it by updating the benchmark's tests, discarding obsolete tasks and integrating new knowledge and challenges in the form of novel test scenarios. This gradual and collective alignment of the benchmark's content is itself a potential risk, since many biases might be introduced in the process, but it is also of great importance for the long-term utility of the benchmark.

# H   Author Statement

The data in this benchmark is either the product of automated generators, or derived from publicly available datasets (ChapterBreak, TriviaQA) which are licensed under Apache 2.0. All the generators were written by the authors except for the generator used to create the Sally-Anne task. That generator is available at `https://github.com/kayburns/tom-qa-dataset` and is currently unlicensed. The Sally-Anne data that our benchmark uses is read from a file that generated from a single run of the the above generator.

The authors of this paper bear all responsibility in case of any violation of rights during the collection of the data or other work, and will take appropriate action when needed, e.g. to remove data with such issues.

