# OpenReview forum: "Beyond Prompts: Dynamic Conversational Benchmarking of Large Language Models"
_NeurIPS.cc/2024/Datasets_and_Benchmarks_Track — NeurIPS 2024 Track Datasets and Benchmarks Poster_

### Official Review · Reviewer_25fo · 2024-07-14
**A dynamic conversational benchmark to evaluate the Long-Term Memory (LTM) and Continual Learning (CL) capabilities of conversational agents.**

**Rating:** 6
**Confidence:** 4
**Correctness:** Yes.
**Clarity:** Yes.

**Review:**

This paper identifies a significant issue with the current suite of benchmarks for LLMs: they do not adequately reflect real-world usage as chat agents, particularly the complexities of a chat environment. To address this, the paper introduces a new benchmark that subjects conversational agents (LLM-based or not) to a series of tests over a single lengthy conversation, evaluating their memory and ability to integrate information.

The paper benchmarks current state-of-the-art large-context LLMs, as well as LLMs equipped with a basic long-term memory (LTM) system. The authors observe that while all scores decline as the benchmark length increases, the scores of agents using an LTM system decline less sharply. This suggests that combining an LLM with a shorter context alongside an LTM system may provide a "focusing" effect for the LLM.

Furthermore, the paper demonstrates that their task-interleaving approach significantly increases the difficulty of the benchmark. Scores varied by up to 1.5 points between benchmarks with interleaved tasks and those with a more traditional isolated task regime. The open-source LLMs particularly struggled with this, potentially revealing differences in the training structures of commercial LLMs.

**Strengths:**

This paper introduces the LTM Benchmark, an automated system designed to evaluate the Long-Term Memory (LTM) and Continual Learning (CL) capabilities of conversational agents. The LTM Benchmark engages agents in a single, prolonged conversation, incorporating multiple tasks and distractions to simulate realistic and meaningful interactions. This approach provides a more comprehensive assessment of an agent’s ability to effectively use and integrate information across extended dialogues. Additionally, this paper demonstrates that this "conversational multitasking" structure significantly degrades the test performance of LLMs, indicating that their real-world capabilities are not fully captured by most contemporary benchmarks. The LTM benchmark is primarily synthetic and can generate conversations of arbitrary length, potentially exceeding the context size of the LLMs being tested.

Strengths:

- An automatic benchmarking system that evaluates an agent by interleaving all tests within a single conversation.
- An initial battery of tests assessing various aspects of LTM and CL, including both the tests themselves and the generators used to create them.
- An extensive evaluation and analysis of results for some of the most advanced language models to date.

**Additional Feedback:**

This work appears somewhat engineering-focused and lacks novelty. I hope the authors can provide more detailed explanations of the contributions and insights.

**Documentation:**

Yes.

**Ethics:**

No.

**Limitations:**

The paper discusses several limitations. For example, the robustness of the agents is assessed based on only three repetitions, which may not be very reliable. Running the benchmark involves significant time and financial costs, making multiple repetitions currently infeasible. Additionally, the episodic memory testing occasionally involves time jumps, necessitating that the agent runs on the same machine as the benchmark system.

**Opportunities For Improvement:**

The main paper lacks details on the generation processes for the synthetic benchmark. It would be beneficial to include this information to ensure the main paper is self-contained.

**Relation To Prior Work:**

Yes.

**Summary And Contributions:**

This paper introduces a dynamic benchmarking system for conversational agents, which evaluates their performance through a single, simulated, and extended user↔agent interaction. During this interaction, multiple tasks are introduced and undertaken concurrently, with regular context switching to interleave the tasks. This creates a realistic testing scenario to assess the agents' Long-Term Memory, Continual Learning, and Information Integration capabilities. Results from both proprietary and open-source Large Language Models indicate that while LLMs generally perform well on single-task interactions, they struggle with the same tasks when interleaved.

---

> ### Author Rebuttal · Authors · 2024-08-13
>
> Thank you very much for your review.
>
> As for the generation processes, there are many details involved and therefore we found ourselves forced to push that material out of the main paper and into the supplementary material, but we agree that it would be optimal for the paper to be self-contained. We will include as much information as we can in the extra page.
>
> In order to mitigate the reliability of the tests, we are conducting additional benchmarks, targeting those configurations with the highest uncertainty. Moreover, we also hope that the cost of running LLMs will decrease over time, which should help make this kind of evaluation more accessible.
>
> In relation to the time jumps, we have implemented a safe interface and fallback. Now, agents can implement their own method for performing the time jumps, or simply wait the expected amount of time. This improved interface should also allow the benchmarking of agents that run remotely.
>
> Finally, in regards to your additional feedback, we believe that the novelty of our approach is in the following:
> - The interleaving of tests. Our isolated testcase demonstration shows that this approach exposes issues with the tested LLMs that the other benchmarks do not capture.
> - The dynamic conversation, which reacts to the agent’s responses and lets it influence the course of the benchmark - see the Restaurant Tests.
> - Our individual tests are mostly synthetic (excluding ChapterBreak), and thus do not rely on external knowledge that may have been scraped and bundled into training datasets.
>
> We do have more of an engineering focus here because the delivery of the data is more complex, and the requirements of this novel evaluation regime inevitably drive us towards tackling the corresponding engineering challenges.

---

### Official Review · Reviewer_zgCD · 2024-07-25
**Good benchmark for conversational agent to a series of tests over a single lengthy conversation.**

**Rating:** 9
**Confidence:** 4
**Correctness:** None
**Clarity:** Well written

**Review:**

1) task-interleaving approach makes the benchmark significantly more difficult, with scores that vary up to 1.5 points between a benchmark with interleaved tasks, and one with a more traditional isolated task regime.
   --> In real scenario, Interleaved tasks may be prevalent. Therefore, this benchmark may be important.

**Strengths:**

1) This benchmark may reflect real usage as chat agents along with the complexities that a chat environment brings.
2) Throughout a single lengthy conversation, this benchmark test the memory of conversation as well as ability for integrating information.

**Additional Feedback:**

None

**Documentation:**

Well done333

**Limitations:**

Nothing special

**Opportunities For Improvement:**

They may do design some scenario for multiple conversations between people.

**Relation To Prior Work:**

Clearly discussed

**Summary And Contributions:**

1) a new benchmark that subjects conversational agents (LLM-based or not) to a series of tests over a single lengthy conversation in order to test their memory and ability to integrate information
2) benchmark current SOTA large-context LLMs, as well as LLMs equipped with a basic LTM system
3) the scores of the agents using an LTM drop less precipitously  while all scores drop as the benchmark length increases,
4) task-interleaving approach makes the benchmark significantly more difficult, with scores that vary up to 1.5 points between a benchmark with interleaved tasks, and one with a more traditional isolated task regime.
5) Since the benchmark has been open sourced on GitHub, many people may update it as the capabilities of LLM and LTM systems mature

---

> ### Author Rebuttal · Authors · 2024-08-13
>
> Thank you for the positive feedback.
>
> We reckon that the multi-user scenario sets interesting challenges in relation to the management of information from multiple sources. We find this scenario interesting because it requires the agent to be highly aware of the contextual cues, and we will include it in the list of future work.
>
> To sketch a scenario using some of the tests that we have presently, we could challenge the agent to give us different versions of shopping lists attributed to different users, even their individual favourite colours. This would require that the agent selects and ignores information based on both context and time.

---

### Official Review · Reviewer_8Qtc · 2024-07-26

**Rating:** 7
**Confidence:** 4
**Correctness:** Yes
**Clarity:** Yes

**Review:**

Weakness:

1. Almost all significant works on Long-Term memory of LLMs are not mentioned or cited, including Memorizing Transformer, MemoryBank, LongMem, MemoryLLM, Memory^3. Please consider to discuss these representative works with your benchmark. In additional to your GoodAI-LTM agent, please also consider to reproduce these LTM framework in the benchmark.

Memorizing transformers. Wu et al. ICLR 2023.

Augmenting language models with long-term memory. Wang et al. NeurIPS 2023.

Enhancing Large Language Models with Long-Term Memory. Zhong et al. AAAI 2024

Towards Self-Updatable Large Language Models. Wang et al. ICML 2024.

Memory3: Language Modeling with Explicit Memory. Yang et al. arxiv. 2024.

2. To the best of my understanding, this benchmark contains only 33 test cases. Is it possible that you can enlarge it to at least over 100? Otherwise, the bias in the evaluation, i.e. the std of 500k span for GPT-4-turbo is 80% of mean score, is too high, making the results completely noisy.

**Strengths:**

1. The LTM of LLMs are an emerging area and there is no effective benchmark on evaluating this. This research direction is of high value and this work encourages the research on that,.

**Additional Feedback:**

None

**Documentation:**

Yes

**Limitations:**

Yes

**Opportunities For Improvement:**

Please use correct bib format and enable hyperlink for cited papers. Do not include Acknowledgements in draft submission.

**Relation To Prior Work:**

Yes

**Summary And Contributions:**

This paper proposed a benchmark to assess the long-term memory capabilities of the LLM agents. It includes 11 scenarios in the requirements of long-term memory and benchmark all SOTA LLMs on that. The conclusion that the LTM framework enables short-context LLM competitive with long-context LLMs.

---

> ### Author Rebuttal · Authors · 2024-08-13
>
> Thank you very much for your time and thorough review.
>
> In relation to the works on LLMs' LTM, we certainly considered at least two of the works that you mentioned in our initial draft, but we ended up not mentioning them for the sake of clarity and conciseness. We had to fit it all within the page limit, and we also thought that readers might get the wrong impression that a SOTA LTM agent was part of our contributions. However, after reading your review it does seem that a brief discussion of those methods would enrich the paper. Additionally, we are currently trying to run the benchmark on some of those agents, and will hopefully have some results that show how SOTA LTM agents react to this more challenging conversational setup.
>
> About the evaluation bias, we tried to do our best within our budget. As disclosed in the paper, we have spent around $5,500 in API costs, and extending the number of tests to 100 would more than triple that cost, since the tests happen as part of the conversation and that would imply more LLM calls with more input tokens. However, we have some suggestions to mitigate this:
>
> 1. We now think that a standard deviation might have been a suboptimal choice in terms of communicating the results. Since the sample is so small (3 tests per task), showing the minimum and maximum score next to the average is probably more informative. In addition, we can show individual results in Figure 5 as dots, instead of bars, to more accurately reflect the sparsity and distribution of the sample.
>
> 2. At this moment, we have been authorised to spend an additional $1,500 on experiments. While it is true that some results have a large standard deviation in relation to the score achieved, this often happens when the scores are really low (0 to 2), and therefore noise is expected. We believe that it could be of greater value to use these credits in increasing the confidence of the results with the highest stakes, decreasing the variance of the configurations that compete for the best positions. In order to do this, we can run a second configuration of the same benchmarks, but with different seeds. Here is a suggestion of eight benchmarks that we could prioritize (those with highest deviations):
>
> - GPT-4 turbo    -  32k - 4.8 ± 3.9
> - GPT-4o         -  32k - 4.6 ± 3.6
> - LTM (Opus)     - 120k - 5.7 ± 3.4
> - LTM (Opus)     - 200k - 6.4 ± 2.9
> - Gemini 1.5 Pro - 200k - 8.0 ± 2.8
> - Claude 3 Opus  - 120k - 7.4 ± 2.7
> - GPT-4o         - 120K - 5.5 ± 2.6
> - Gemini 1.5 Pro - 500k - 5.3 ± 2.6
>
> We can run the additional benchmarks in this order, until using up the budget, and show them with a mark to denote the running of additional tests.
>
> Please let us know your thoughts about these points. Any suggestions are also welcome. We would also like to ask for clarification in regards to the correct bib format. We used the LaTeX style provided and checked the style guidelines, and still we cannot identify the issue.

---

> > ### Author Rebuttal · Authors · 2024-08-14
> >
> > We are currently executing the additional runs that we mentioned in the previous message, and we have identified MemoryBank as a possible other LTM agent to evaluate.
> >
> > Additionally, we have been discussing the issue of the data scarcity and we have arrived to interesting conclusions that we think might help solve the problem. It all comes down to the assumptions on the nature of the tests.
> >
> > Currently, the results from different repetitions of the same test scenario are grouped together, and a mean and standard deviation are computed from them. Finally, all means and standard deviations are added up to reach the global score. This procedure already assumes some level of independence among the test repetitions, but let’s delve into that.
> >
> > Because all test repetitions take place within the same conversation, they inevitably have some degree of dependency. However, if we tell all factors apart, for each repetition we have needles (relevant information) and distractors (filler messages and needles from other test scenarios), and if we look at the data from this perspective, the results collected from different repetitions of the same test scenario can be said to be statistically independent. In other words, they may have different contexts and therefore different levels of difficulty, but they all evaluate the same aspect of the agent’s LTM skills.
> >
> > Now, this still leaves us with only 3 samples per scenario, but we can leverage the assumption of independence to augment our results by generating a distribution of possible outcomes. Instead of simply computing group-level standard deviations and aggregating them up, we can uniformly sample one result per test scenario and add them up to obtain a possible global score. If we do this enough times (e.g. 1000), we will have a more faithful distribution of the agent’s overall score on the benchmark, which we can show in box plots and compute a standard deviation from.
> >
> > To this message we have attached a document in which we show the points resulting from this method, and present an alternative to Figure 5 using box plots. We have confirmed that the sampling method produces the same average results as the previous method, and the standard deviations are lower but consistent (convergence values are the same among different sample sizes). We believe that this method is both reasonable and more appropriate to show the results, and it should not lead to misunderstandings given that an appropriate explanation is provided.

---

> ### Author Rebuttal · Authors · 2024-08-26
>
> To follow up on the testing of other LTM methods, we have attempted to benchmark a number of the implementations that you have listed in your review.
>
> There are two types of LTM systems in the literature, those that rely on a customised model, (often Llama based) and those that do not. Customised models like in the case of MemoryLLM seemingly have a disadvantage in that they are effectively capability-locked to their base LLM.
>
> We have adapted implementations for two LLM agnostic systems: MemGPT and MemoryBank. Both MemoryBank and MemGPT had some challenges. One common issue was that neither system had much respect for context limits. We tried to correct for this, but there still may be cases where the context for a given request exceeds the nominal limits that we place on the context size.
>
> MemoryLLM is based on Llama2. When adapting it to the benchmark, we found out that the released model was not instruction tuned, which was also confirmed by the author. As a result, the continuous nature of our benchmark often breaks the token generation of the model after ~5 prompts.
>
> Given our current cost constraints, we used GPT-4o-mini to run both MemGPT and MemoryBank. We have also included benchmarks for our LTM agent using 4o-mini as well as a 4o-mini run itself.
>
> MemGPT uses a variety of memory types, such as core (always visible to the agent) and archival (searchable by the agent) memories. The agent is empowered to use functions to search for, add, or remove memories. MemGPT performs consistently below our LTM baseline, but still follows the LTM trend of being still able to answer questions even in the 500k benchmarks.
>
> MemoryBank, an LTM system that underpins Silicon Friend, was retooled for use on our benchmark requiring minior alterations. It uses Llamaindex to store all memories as documents in a simple vector store, these documents are written to and retrieved from and are composed of single user query and agent response pairs. While we were able to run it on our benchmark the results were relatively low due to the limited context size of four thousand (excluding retrieved memories), only retrieving the five most relevant documents, and utilising ChatGPT 4o Mini. Promisingly however, there was no falloff in the score as the benchmark context size increased, again demonstrating long-term memory's positive impact.
>
> As mentioned above in our previous rebuttal, we have devised a method of interpreting the results and obtaining score distributions. From those, we have produced new figures and updated the standard deviations according to the new method described in the rebuttal above. In the table, the underlined entries are those that have been tested on additional data.

---

> > ### Comment · Reviewer_8Qtc · 2024-08-28
> >
> > Thanks for the significant additional experiments with GPT-4o-mini-MemoryGPT and GPT-4o-mini-MemoryBank. It is very surprising that all these two methods do not outperform vanilla LTM GPT-4-turbo. These results are significant and may demonstrate the under-explored issue of the current research on LTM-LLMs. I increased the score to 7 and incline to acceptance.

---

### Decision · Program_Chairs · 2024-09-26

**Decision:**

Accept (Poster)

**Comment:**

This paper introduces a dynamic benchmarking system,  designed to evaluate conversational agents' Long-Term Memory (LTM) and Continual Learning (CL) capabilities. It provides a realistic, multitasking testing environment for conversational agents.
Demonstrates that LLMs with a Long-Term Memory system maintain performance better in extended tasks.
Highlights the significant impact of task interleaving on performance, reflecting real-world usage scenarios.